# Turn-timing in conversations between autistic adults: Typical short-gap transitions are preferred, but not achieved instantly

**Simon Wehrle**[1]*, **Francesco Cangemi**[1], **Alicia Janz**[1], **Kai Vogeley**[2,3], **Martine Grice**[1]

**1** Institut für Linguistik-Phonetik, University of Cologne, Cologne, Germany, **2** Department of Psychiatry, University Hospital Cologne, Medical Faculty, University of Cologne, Cologne, Germany, **3** Cognitive Neuroscience, Institute of Neuroscience and Medicine (INM-3), Research Center Jülich, Jülich, Germany

* simon.wehrle@uni-koeln.de

**Data Availability Statement:** All data and scripts are available in the OSF repository at https://osf.io/v5pn4/. The data folder contains one csv file with the experimental data and one csv file with subject

## Abstract

The organisation of who speaks when in conversation is perhaps the most fundamental aspect of human communication. Research on a wide variety of groups of speakers has revealed a seemingly universal preference for between-speaker transitions consisting of very short silent gaps. Previous research on conversational turn-taking in Autism Spectrum Disorder (ASD) consists of only a handful of studies, most of which are limited in scope and based on the non-spontaneous speech of children and adolescents. No previous studies have investigated dialogues between autistic adults. We analysed the conversational turn-taking behaviour of 28 adult native German speakers in two groups of dyads, in which both interlocutors either did or did not have a diagnosis of ASD. We found no clear difference in turn-timing between the ASD and the control group overall, with both groups showing the same preference for very short silent-gap transitions that has been described for many other groups of speakers in the past. We did, however, find a clear difference between groups specifically in the earliest stages of dialogue, where ASD dyads produced considerably longer silent gaps than controls. We discuss our findings in the context of the previous literature, the implications of diverging behaviour specifically in the early stages of conversation, and the general importance of studying the neglected aspect of interactions between autistic adults.

## 1. Introduction

Turn-taking is in essence a form of cooperative interaction. Humans engage in many temporally coordinated collaborative activities besides spoken interaction, such as manual labor, dancing or music-making (see e.g. [1]). Similarly, communicative turn-taking, in either the vocal or gestural modality, is not limited to humans. Many different species from different taxa perform tightly synchronised and regulated communicative interactions (e.g. [2]). Human turn-taking in conversation, however, is a unique and remarkable phenomenon. It is not only executed with split-second precision and flexibility and involves the parallel

data. The main folder contains an RMarkdown file (in .rmd and .html formats) in which the entire paper is reproduced with code chunks that were used to produce all plots and perform all modelling adjacent to the relevant portions of the paper.

**Funding:** Funding was provided by the Deutsche Forschungsgemeinschaft (CRC1252 Prominence in Language, Project A02) to MG and KV (http://www.dfg.de) and by the Studienstiftung des Deutschen Volkes (Promotionsstipendium) to SW (http://www.studienstiftung.de). The funders had no role in study design, data collection and analysis, decision to publish, or preparation of the manuscript.

**Competing interests:** The authors have declared that no competing interests exist.

prediction, planning and production of utterances which are improvised yet rich with meaning, but it is also the key means through which human language, and to a considerable extent human culture, are learned and transmitted (cf. [3]).

Most turns in spoken conversation are short and most transitions between turns consist of very short gaps between speakers [4], which are preferred to other possible kinds of transition such as longer gaps or overlaps (two speakers talking at once).

A modal value of around 200 milliseconds of silence between speakers has been shown for a wide range of languages and speakers, with only minor language-specific variations [5–8].

As successful and rapid turn-timing crucially relies on socio-communicative abilities such as pragmatic language skills [4, 9, 10], which are typically thought to be impaired in individuals on the autism spectrum, delayed or otherwise divergent patterns of turn-timing in this population might plausibly be predicted.

However, there is only very limited quantitative research on turn-timing in Autism Spectrum Disorder (ASD) to date, and none whatsoever on turn-timing in conversations between autistic adults. The limited experimental evidence available seems to point to a general tendency for longer silent gaps in conversations involving autistic participants (see Section 4.2), although it is not clear to which extent this trend can be expected to apply to (semi-)spontaneous conversations between autistic adults, which are investigated in this study for the first time.

We present an analysis of turn-taking strategies in pairs of German adults, where both interlocutors either did or did not have a diagnosis of ASD (in so-called disposition-matched dyads). When considering the dialogue as a whole, we found no clear differences in turn-timing between the ASD and the control (CTR) group. However, closer inspection reveals that, compared to CTR dyads, autistic dyads produced longer gaps between turns specifically in the earliest stages of dialogue. We discuss the implications of these results and relate them to previous research on autism and to the notion of seemingly universal patterns of turn-timing in spoken dialogue.

## 2. Materials and methods

This section provides details on 1) the subjects that participated in this study, 2) the materials and set-up used and 3) the data set and methods of analysis.

### 2.1. Subjects

We recorded 28 monolingual native speakers of German (14 ASD, 14 CTR) engaged in semi-spontaneous conversation. Participants were grouped into disposition-matched dyads (7 ASD–ASD, 7 CTR–CTR). Participants from the ASD group had all been diagnosed with autism (corresponding to ICD-10: F84.0; see [11]) or Asperger syndrome (ICD-10: F84.5) and were recruited in the Autism Outpatient Clinic at the Department of Psychiatry of the University of Cologne (Germany). The key diagnostic criteria described in the ICD-10 (International Statistical Classification of Diseases and Related Health Problems) are 1) unusual ("impaired") social interaction and communication and 2) a restricted repertoire of activities and interests. As part of a systematic assessment implemented in the clinic, diagnoses were made independently by two different specialized clinicians corresponding to ICD-10 criteria, and supplemented by an extensive neuropsychological assessment.

Subjects from the ASD group were first recorded and described by [12, 13] (performing different tasks). Participants from the control group were recruited from the general population specifically for this study and were paid 10 EUR each for participation.

All participants completed the German version of the Autism-Spectrum Quotient (AQ) questionnaire, an instrument developed by [14] to measure autistic traits in adults with normal intelligence. AQ scores range from 0 to 50, with higher scores indicating more autistic traits. An AQ score of 32 or above is commonly interpreted as a clinical threshold for ASD [14, 15]. All subjects in our ASD group scored above the suggested threshold of 32 points and all subjects in our CTR group scored below the same threshold. All participants also completed the *Wortschatztest WST* [16], a standardised, receptive German vocabulary test that exhibits high correlation not only to verbal intelligence, but also to general intelligence [17].

Although participants from the CTR group were matched as closely as possible to the ASD group for age, verbal IQ (intelligence quotient) and gender, some minor differences remained. Participants from the ASD group were on average slightly older (mean = 44; range: 31–55) than participants from the CTR group (mean = 37; range: 29–54). However, there was extensive overlap between groups and, moreover, there is no a priori reason to assume that such a relatively small difference in this particular age range would act as a confound in group comparisons.

Further, the ASD group had a slightly higher average verbal IQ score (mean = 118; range: 101–143) than the CTR group (mean = 106; range: 99–118). Again, there was considerable overlap between groups. We have no reason to assume that this difference should have a meaningful impact on results.

The gender ratio was similar, but not identical across groups. The ASD group contained 4 females and 10 males, whereas the CTR group contained 3 females and 11 males. This entails that dialogues took place in the ASD group between 1 all-female dyad, 2 mixed dyads and 4 all-male dyads, but in the CTR group between 3 mixed dyads and 4 all-male dyads (i.e. no all-female dyad). Using Bayesian modelling (see further information in Section 2.3), we were able to confirm that these small differences between groups did not, however, have any effect on the analyses presented here. We will therefore disregard gender as a factor in reporting the experimental results. (We used a Gaussian model with FLOOR TRANSFER OFFSET as the dependent variable, GENDER COMBINATION (all-female/all-male/mixed) as a fixed factor and DYAD as a random factor, and found no robust differences between any of the groups—more details in the accompanying *OSF* repository at https://osf.io/v5pn4/).

Most importantly, there was a clear difference in AQ scores between groups, with a far higher average score in the ASD group (mean = 41.9; range = 35–46) than in the CTR group (mean = 16.1; range: 11–26) and no overlap at all between subjects from the two groups. Bayesian modelling provides unambiguous evidence for the group difference in AQ scores, and also confirms that the differences in age and verbal IQ are small but robust. Table 1 shows summary statistics for gender, age, verbal IQ and AQ. (We used Poisson models with AGE/VERBAL IQ/AQ as the respective dependent variable, and GROUP (ASD/CTR) as the independent variable in all cases).

All aspects of the study were approved by the local ethics committee of the Medical Faculty at the University of Cologne and were performed in accordance with the ethical standards

**Table 1. Subject information by group.**

| Gender (n) | | Age | | Verbal IQ | | AQ | |
|---|---|---|---|---|---|---|---|
| female | male | Mean | SD | Mean | SD | Mean | SD |
| ASD 4 | 10 | 43.6 | 6.7 | 118.1 | 12.0 | 41.9 | 3.1 |
| CTR 3 | 11 | 36.5 | 7.6 | 105.8 | 5.8 | 16.1 | 4.5 |

SD = standard deviation.

laid down in the 1964 Declaration of Helsinki and its later amendments. All subjects gave their written informed consent before participating in the experiment.

## 2.2. Materials and procedure

We used Map Tasks to elicit semi-spontaneous speech. The Map Task paradigm was introduced by [18] and has widely been used in speech research for over 30 years (see [19] for an influential article describing a corpus of Map Task speech).

Materials consisted of pairs of simple maps. Each map contained 9 landmark items in the form of small pictures (materials adapted from [20]). Only one of the two participants in each Map Task (the instruction giver) had a route printed on their map. The experimental task was for the instruction follower to transfer this route to their own map by exchanging information with the instruction giver.

In each map, some landmarks were either missing, duplicated and/or replaced with a different landmark, compared to the interlocutor's map. This was the case for 2 landmarks per map. Those items that differed between maps will hereafter be called Mismatches; items that were the same on both maps will be called Matches. During annotation, we marked the portion of dialogue in which the first Mismatch was discussed by participants and used it to divide all dialogues up into three epochs, i.e., before detection, during discussion, and after resolution of the first Mismatch (see Section 3.2 for further details). An example of maps used in this study is shown in Fig 1, with Mismatches highlighted using red circles. All dyads received the same two pairs of maps.

Participants in the study first filled in a number of forms and the questionnaires listed in Section 2.1, then received written instructions for the task and finally entered a recording booth. They then received one map each (only one of which featured the route from start to finish). During this entire process, an opaque screen was placed between participants, meaning they could not establish visual contact and had to solve the task by means of verbal communication alone. We chose to restrict conversations (and the subsequent analysis) to the spoken modality as we were not equipped to perform in-depth analyses of multi-modal interaction at the time of recording. The roles of instruction giver and instruction follower were assigned randomly. Upon completion of the first task, subjects received a new set of maps and their roles were switched. The task ended once the second Map Task was completed. As participants were naive to the purpose of the study, they did not know at the outset that their maps differed in some crucial regards.

All conversations were recorded in a sound-proof booth at the Department of Phonetics, University of Cologne. We used two head-mounted microphones (AKG C420L) connected through an audio-interface (PreSonus AudioBox 22VSL) to a PC running Adobe Audition. The sample rate was 44100 Hz (16 bit). Recordings were transcribed orthographically and divided into interpausal units (IPUs) with a minimum pause length of 200 ms.

We only included recorded dialogue from the start to the end of each task in all analyses, in order to achieve a greater degree of comparability regarding conversational context and content. The total duration of speech material is 4 hours and 44 minutes. The mean dialogue duration is 20 minutes and 19 seconds (SD = 12'32").

Fig 2 shows an example excerpt of Map Task dialogue from one of the ASD dyads, transcribed following GAT conventions [21, 22]. Two examples of turn transitions are highlighted in bold, one following the introduction of a matching landmark—"heller Diamant", line 15/16—and one following the introduction of a mismatching landmark—"goldene Moschee", line 21/22. Note that the turn transitions highlighted here are considerably longer than average transitions between turns.

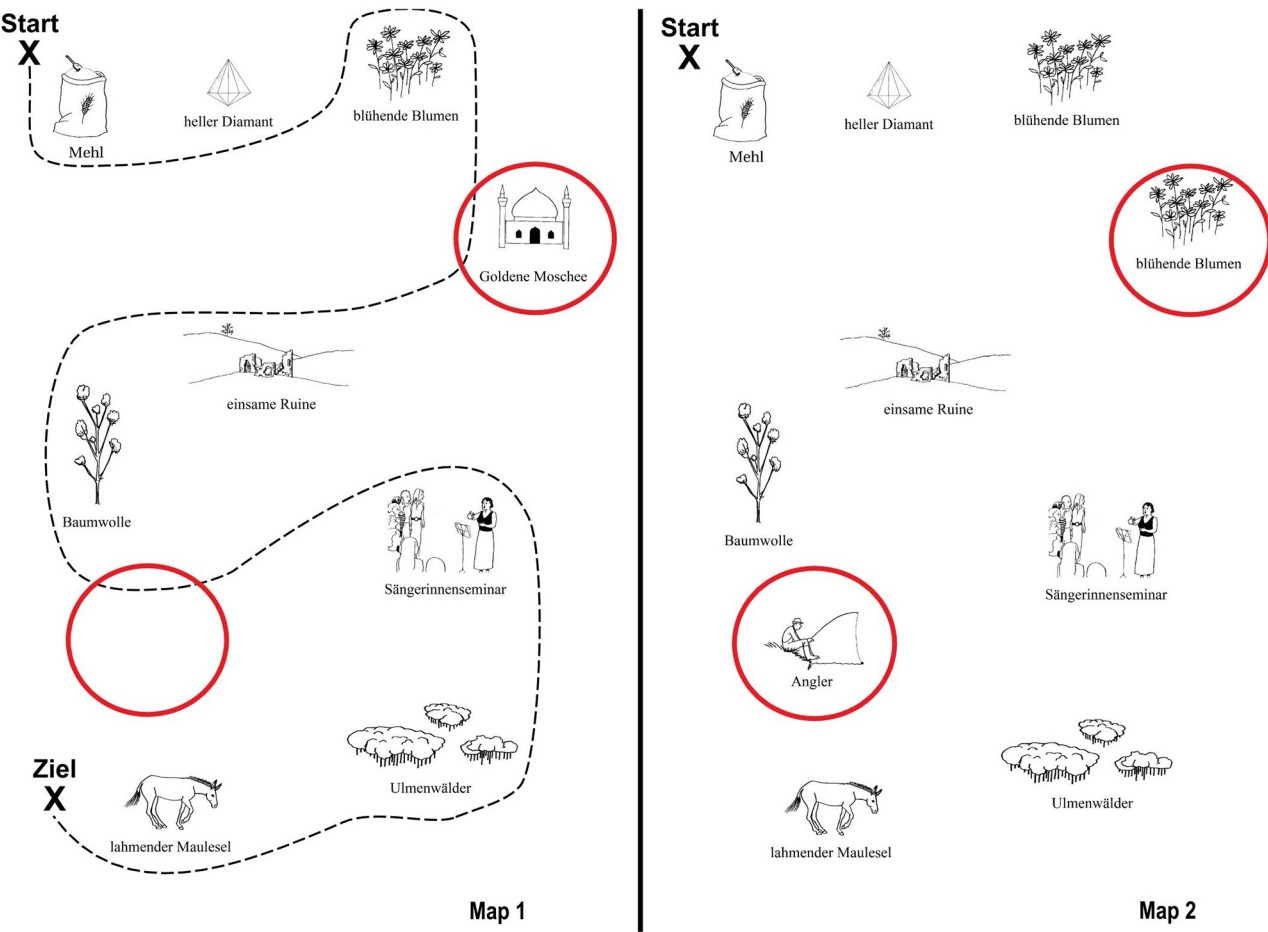

**Fig 1. Example of map task materials.** The instruction giver's map, with a route leading from "Start" (top left) to "Ziel" (finish; bottom left), is in the left panel. Mismatches between maps are highlighted with red circles.

We chose the Map Task paradigm for the current investigation as it provides us with predominantly spontaneous speech data that can, however, still be controlled along a number of key parameters, such as lexical items (via the names of landmarks on a map) and communicative obstacles (such as the introduction of mismatching landmarks between maps). While the elicited dialogues are not fully free or spontaneous, the Map Task seemed to us a good choice in the context of comparing autistic and non-autistic dyads, since the constraints involved in the task serve to reduce a potentially particularly high degree of variability across the autism spectrum in terms of social motivation, interest in a given topic, and the adherence to social conventions.

### 2.3. Data and analysis

Our data set contains 18332 IPUs in total (inter-pausal units; here defined as speech separated by at least 200 milliseconds of silence). For an analysis of turn-taking, not these units of speech in themselves are of primary interest, but rather the points of transition between them. Our data set contains 5668 such transitions overall. There are fewer turn transitions than IPUs because most of the latter were followed by another IPU from the same speaker; i.e. separated by within-speaker pauses rather than between-speaker gaps.

```
13    S1:    (-) okay   du   gehst unter dem mehl durch
                  okay   you    go  under the flour through
                  'okay you pass below the flour'

14    S2:    j  ja
             y  yes

15    S1:    °h in richtung  heller diamant
                in direction bright diamond
                'towards the bright diamond'
```

→ **1738 ms gap (Match)**

```
16    S2:    °hhh okay

17    S1:    dann gehst du o:bm    über die blu blühenden  blumen und zwar
wirklich über   die blumen
                then  go  you above   over the flo-blossoming flowers and indeed
really   above  the flowers
                'then you go above the blossoming flowers, really above the flowers'

18           nicht  über  die (.) äh über  die buchstaben
             not     above the     uh above the letters
             'not above the writing'

19    S2:    achso    über  die blühenden  blumen  ja
             oh right above the blossoming flowers yes
             'I see, above the blossoming flowers, yes'

20    S1:    oben drüber [dann ge]hst du runter °hh [gehst] an der goldenen moschee
vorbei
                up    above [then go]   you down        [go]   at the golden    mosque
past
                'above it, then you go down, you go past the golden mosque'

21    S2:                 [ja]                 [runter]
                          [yes]                [down]
```

→ **971 ms gap (Mismatch)**

```
22           °hhh goldene moschee
                 golden  mosque

23    S1:    (-) ja °hhh du  gehst [weiter] runter an der einsamen
                 yes      you go  [further] down  at the  lonely
             'yes you go further down, past the lonely'

24    S2:                              [hmm]

25           ich hab aber hier keine goldene moschee
             i   have but here no     golden  mosque
             'but I don't have a golden mosque here'

26    S1:    (-) du  hast keine goldene moschee
                 you have no    golden  mosque

27    S2:    nein
             no

28    S1:    °hh dann hast du  eine andere    landkarte
                then have you  a    different map
             'then you've got a different map'
```

**Fig 2. Example excerpt of a GAT transcription.** Two turn transitions (following newly introduced landmarks) are highlighted in bold (lines 15/16; 21/22).

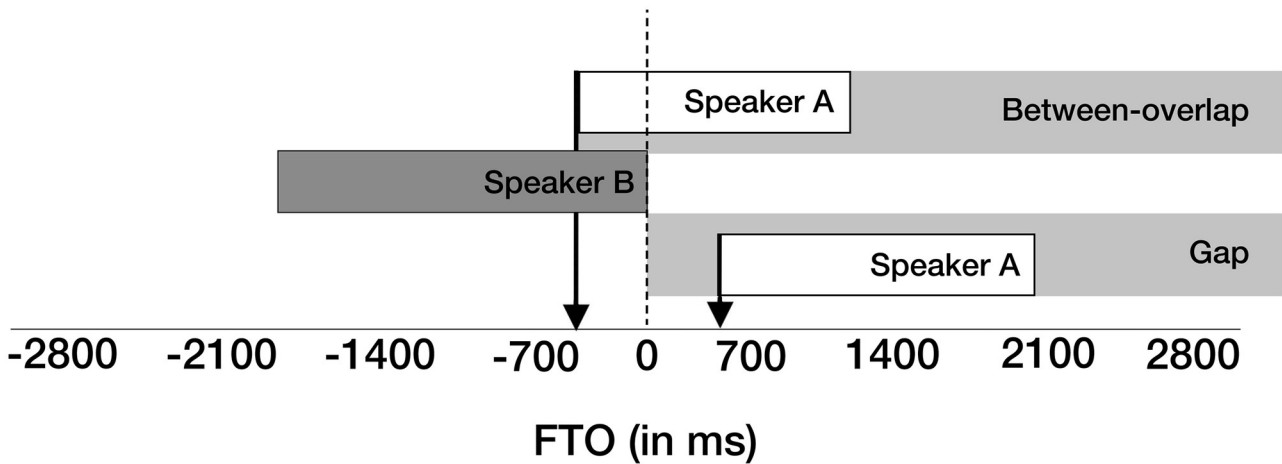

**Fig 3. Categories and measurement of turn transitions.** "Gaps" are silent intervals between turn transitions; "between-overlaps" are turn transitions composed of overlapping speech from both interlocutors. Gaps are represented with positive Floor Transfer Offset (FTO) values (see right arrow for an FTO value of about +600 ms); overlaps are represented with negative FTO values (see left arrow for an FTO value of about -600 ms).

Start and end points of all transitions were precisely labelled by hand following an automatic first-pass segmentation of recordings into silent and non-silent intervals using *Praat* (version 6.1.09) [23]. We aimed to broadly follow the methodology of [24] (which in turn builds on [5]) for the continuous analysis of turn-timing, in order to facilitate comparison of our results to previous work. Following this approach, we counted audible in-breaths, clicks and similar noises as part of silent intervals. Filled pauses such as <uhm>, on the other hand, were annotated as being part of non-silent utterances. Thus, we followed the approach of essentially analysing turn-timing from a linguistic, rather than a purely acoustic perspective (which would incidentally not solve the problem of analysts having to subjectively determine thresholds for what is considered "silence").

Following [24], we categorised all turn transitions as being either "gaps" or "between-overlaps". Turn transitions were analysed using the measure of Floor Transfer Offset (FTO), in which positive values represent gaps and negative values represent overlaps between speakers. Fig 3 gives a schematic representation.

The related category of "within-overlaps" refers to cases in which one speaker's ongoing turn contains a period of overlap with speech from the interlocutor, but is not followed by a change of speaker [24]. In other words, these are situations where Speaker A has started and continues speaking, Speaker B then produces a simultaneous utterance (e.g. "yes"), but then falls silent again, with only Speaker A continuing to speak (see S1 Fig). This does not entail a floor transfer from one speaker to another and such cases did therefore not enter into the analysis of turn-timing that is the main focus of this paper. Briefly, distribution and characteristics of within-overlaps were very similar in ASD and CTR conversations: they were typically very short (around 380 ms) and contained backchannel tokens (listener signals such as "mmhm" or "yeah") in around 70% of cases, for both groups (and e.g. answers to tag questions, or longer utterances, in the other cases; see e.g. [25], Part IV, for further details on the analysis of backchannels).

Of the 5668 transitions in the data set, 3418 were silent gaps (60.3%), 1326 were between-overlaps (23.3%) and 924 were within-overlaps (16.3%). After the exclusion of within-overlaps, 4744 transitions remained for the analysis of turn-timing. Of these, 72% were gaps, 28% (between-)overlaps (further information in S1 Table).

In reporting the results of this essentially exploratory study, we emphasise detailed description and data visualisation [26, 27] along with an in-depth analysis of dyad-specific behaviour [28–31]. We use Bayesian inference to corroborate our findings, but consider descriptive, exploratory analysis to be at the heart of this work. Therefore, we report the most essential elements of the Bayesian models used in the paper itself, but for all further details, as well as all data frames, scripts and codes used to generate the analyses and plots in this paper, we refer the reader to the accompanying *OSF* repository at https://osf.io/v5pn4.

We chose to use Bayesian rather than frequentist statistics for a number of reasons. First, given the limited sample size of the study at hand as well as the scant previous research on the topic, we deem presenting our results and analysis as exploratory, rather than confirmatory, as the best option. Bayesian inference is particularly well suited to studies with a limited sample size, as this limitation can be directly reflected in the model output (e.g. in the form of larger credible intervals and a lower posterior probability). The method gives outcomes based on the data at hand, the chosen model and the specified prior assumptions. Compared to frequentist inference, it is therefore, when properly applied, more conservative, but also more robust and transparent than frequentist approaches [32–35]. Second, Bayesian inference is rapidly increasing in popularity in linguistics and many other fields. This is due in part to practical reasons, as recent statistical software, tutorials and packages have made the application of Bayesian multilevel modelling increasingly straightforward and at the same time considerably more robust and flexible than the frequentist alternatives [36]. Additionally, Bayesian methods seem to be much more closely aligned with common human intuitions and ways of reasoning about the interpretation of statistical tests in general and the notion of significance in particular [35, 37, 38].

For the main analysis, we tested for group differences in FTO values as well as the interaction of group with part of dialogue (see Section 3.2 for details). All models included random intercepts for dyads. We used Bayesian multilevel linear models implemented in the modelling language *Stan* (version 2.29) [39] via the package *brms* (version 2.16.3) [40] for the statistical computing language *R* (version 4.1.2) [41], which we used in the software *RStudio* (version 2021.09.1) [42]. We report expected values ($\beta$) under the posterior distribution and their 95% credible intervals (CIs). We also report the posterior probability that a difference $\delta$ is greater than zero. In essence, a 95% CI represents the range within which we expect an effect to fall with a probability of 95%. We used regularising weakly informative priors for all models [33, 43]; all priors were centred at zero and distributions were chosen according to relevant results in the previous literature (e.g. the feasible range of FTO values). We performed posterior predictive checks with the packages *brms* (version 2.16.3) [40] and *bayesplot* (version 1.8.1) [44] in order to verify that the priors were suited to the data set. Unless otherwise specified, four sampling chains ran for 4000 iterations with a warm-up period of 2000 iterations for each model.

Besides the packages for Bayesian modelling, we made extensive use of the packages included in the *tidyverse* collection for performing data import, tidying, manipulation, visualisation and programming [45].

## 3. Results

The following results are presented using the measure of Floor Transfer Offset (FTO), which allows for a continuous representation of both gaps and overlaps along the same dimension by representing gaps between speakers as positive values and overlaps as negative values.

## 3.1. Overall results

Fig 4 shows turn-timing values by group. Visual inspection alone makes it clear that values are very similar across groups. Overall, the ASD group had slightly higher FTO values, with a mean of 317 ms (SD: 599) and a median of 205 ms, compared to the CTR group with a mean of 238 ms (SD: 555) and a median of 175 ms.

Assuming 100-millisecond bins, both the ASD and the CTR group have a modal FTO value of 200 ms. In this regard, our study essentially replicates a number of previous findings on turn-timing from [6] onwards. S2 Fig presents histograms using 100-millisecond bins and is directly modelled after the histograms presented in [24].

Fig 5 presents FTO values by dyad. The plot clearly shows that distributions are extremely similar across dyads. Note, for instance, that the dashed line at the 200 ms mark (indicating very short gaps) runs close to the distributional peak of all dyads from both groups. Assuming a bin width of 100 milliseconds, 11 out of all 14 dyads produced a modal value of 200 ms (with the modes of the remaining dyads not deviating by more than 100 ms). Mean FTO values ranged from 137 ms to 503 ms across dyads. The group-level tendency towards slightly higher FTO values in the ASD group is reflected in the fact that four out of the five highest mean FTO values were produced by ASD dyads and four out of the five lowest mean values were produced by CTR dyads.

To corroborate the representativeness of group-level results, we tested whether any single dyad had a decisive influence on the group level patterns by successively omitting individual dyads and rerunning the group-level analysis, but found this not to be the case.

**3.1.1. Statistical analysis.**   We used a Gaussian model with FTO as the dependent variable, GROUP (ASD/CTR) as a fixed factor and DYAD as a random factor. The group difference in the model is reported with ASD as the reference level. Mean $\delta$ = -72, indicating a trend towards

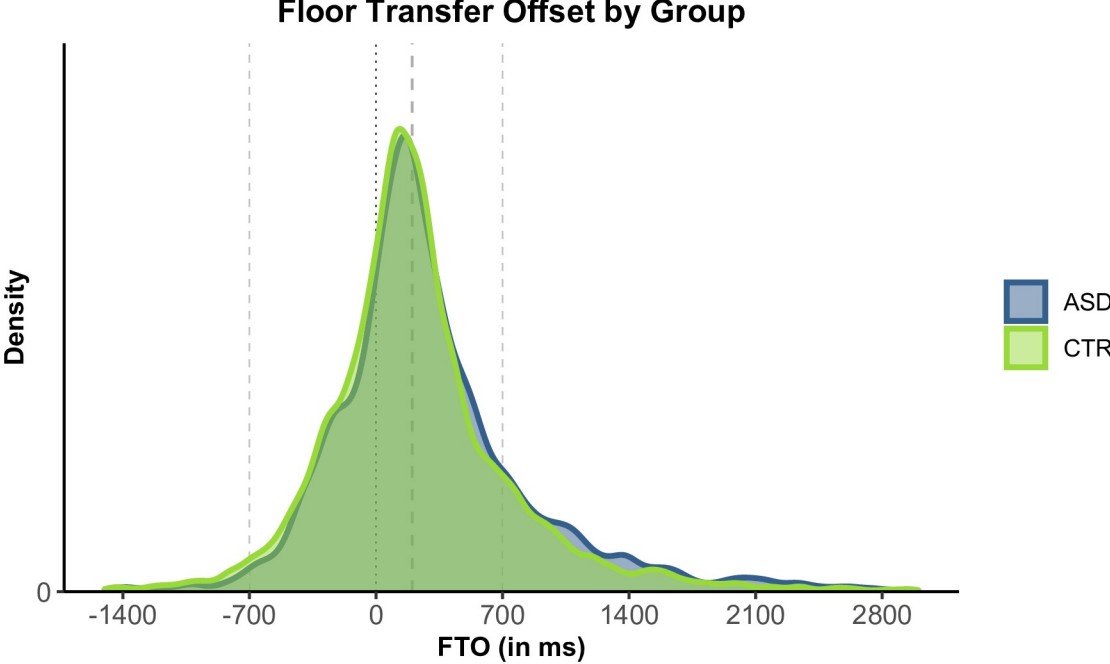

**Fig 4. Floor Transfer Offset (FTO) values by group.** Positive values represent gaps; negative values represent overlaps. ASD group in blue, CTR group in green. The dotted line indicates the value of 0 ms FTO, representing no-gap-no-overlap transitions. Dashed lines indicate the values of +200 ms (expected for typical transitions) and +/-700 ms FTO (unusually long transitions).

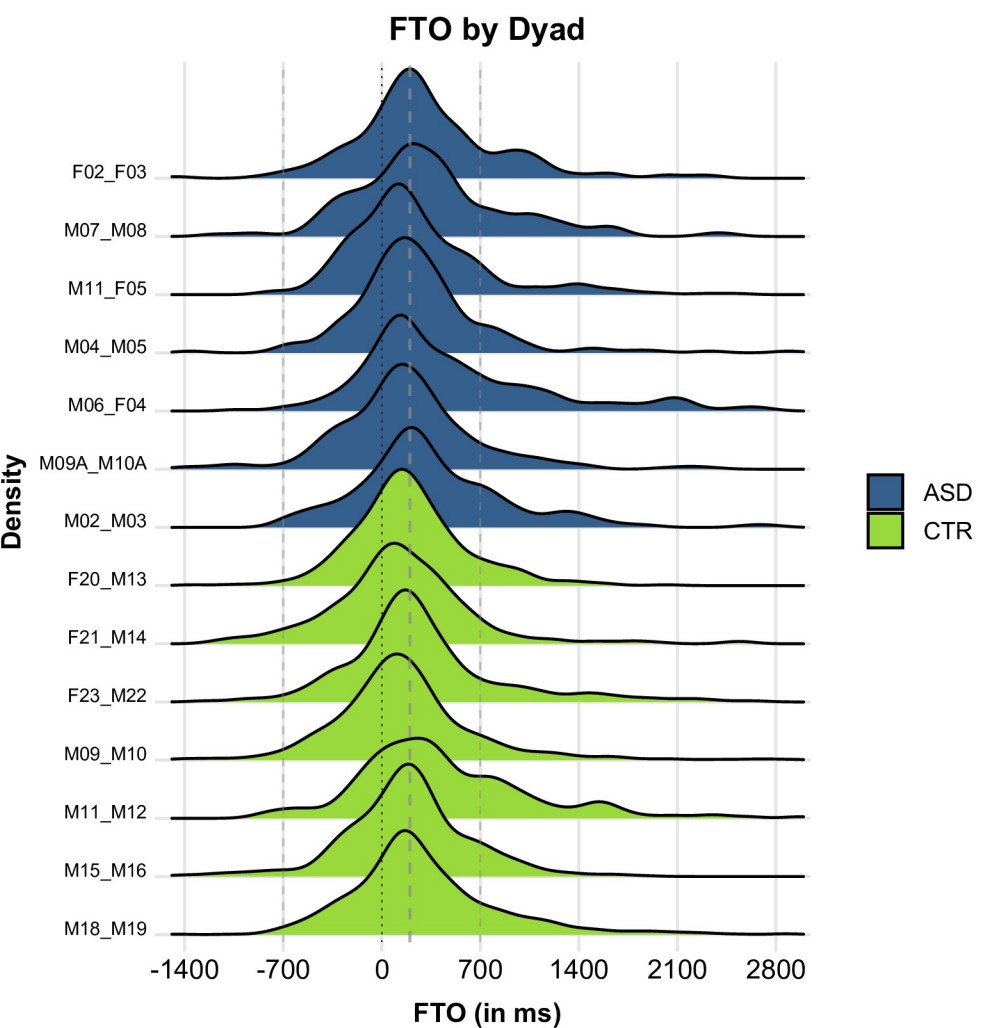

**Fig 5. FTO values by dyad.** Positive values represent gaps; negative values represent overlaps. ASD group in blue, CTR group in green.

slightly lower FTO values in the CTR group. However, the 95% CI [-174, 32] includes zero by some margin and the posterior probability $P(\delta > 0) = 0.88$ is relatively low. The model therefore shows a trend towards higher FTO values (i.e. longer gaps) in autistic dyads, but does not suggest this to be a robust difference between groups.

### 3.2. Results by dialogue stage

Although the turn-timing behaviours of the ASD and the CTR group were quite similar overall, some clear differences between groups are revealed when we do not only consider FTO results across the dialogue as a whole, but also compare early with later dialogue stages. We here use detection of the first Mismatch in the first Map Task as a cut-off point (see Section 2.2): all dialogue preceding detection is counted as being part of the beginning of the conversation, all dialogue following detection as the remainder of the conversation (more details in section 3.2.1).

Fig 6 shows FTO values by group and dialogue stage. While for most of the dialogue autistic dyads performed turn-timing essentially equivalent to that of non-autistic dyads, we can see

**Floor Transfer Offset**

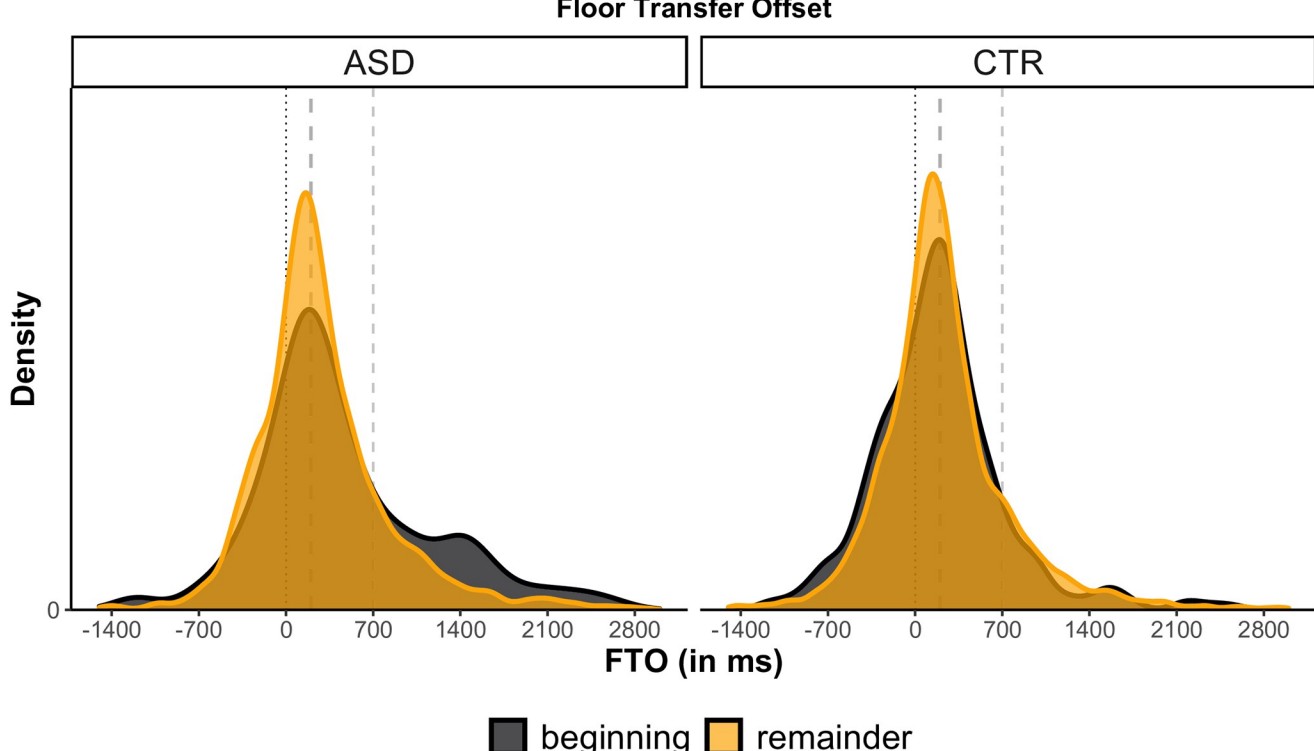

**Fig 6. FTO values by group and dialogue stage.** The black curve represents the beginning of the dialogue (until detection of the first Mismatch); the orange curve represents the remainder of the dialogue (after detection of the first Mismatch). Positive values represent gaps; negative values represent overlaps. ASD group on the left, CTR group on the right.

that, crucially, they did not arrive at this timing instantly. In fact, during the first few minutes of dialogue, before the first Mismatch in the Map Task was detected (2 minutes or 10% of overall duration into the task on average), FTO values for the ASD group were far higher (mean = 511 ms; SD = 799) than in the remainder of the dialogue (mean = 299 ms; SD = 576). These values indicate considerably longer silent gaps between ASD dyads early in the task. The CTR group instead only shows a slight change, and in the opposite direction, with shorter gaps (and slightly more overlaps) in the beginning of the dialogue (mean = 191 ms; SD = 530) compared to the remainder (mean = 243 ms; SD = 558). This interaction signifies that the turn-timing behaviour of the CTR and the ASD group differed considerably in the beginning of conversations ($\delta$ = 320 ms), but not at later stages ($\delta$ = 56 ms).

Fig 7 presents FTO values by dialogue stage and dyad, with CTR dyads in the top half of the plot and ASD dyads in the bottom half. We can see that for most (but not all) CTR dyads, FTO values were essentially the same for early and later stages of dialogue. For most (but not all) ASD dyads, on the other hand, there was a lot of variability in the early stages of dialogue, mostly (but not only) in the direction of longer gaps. This variability disappeared after the initial stages, as the dyads seemed to settle into a temporally stable turn-taking style that is virtually indistinguishable from that of CTR dyads.

**3.2.1. Corroboration of dialogue stage effect.** In all the above analyses, we used *detection* (i.e. first mention) rather than *resolution* of the first Mismatch (i.e. the time when interlocutors finished discussing the first Mismatch and moved on to the remainder of the task) as a cut-off point for the early stages of dialogue. There are two main reasons for this choice. First, we have

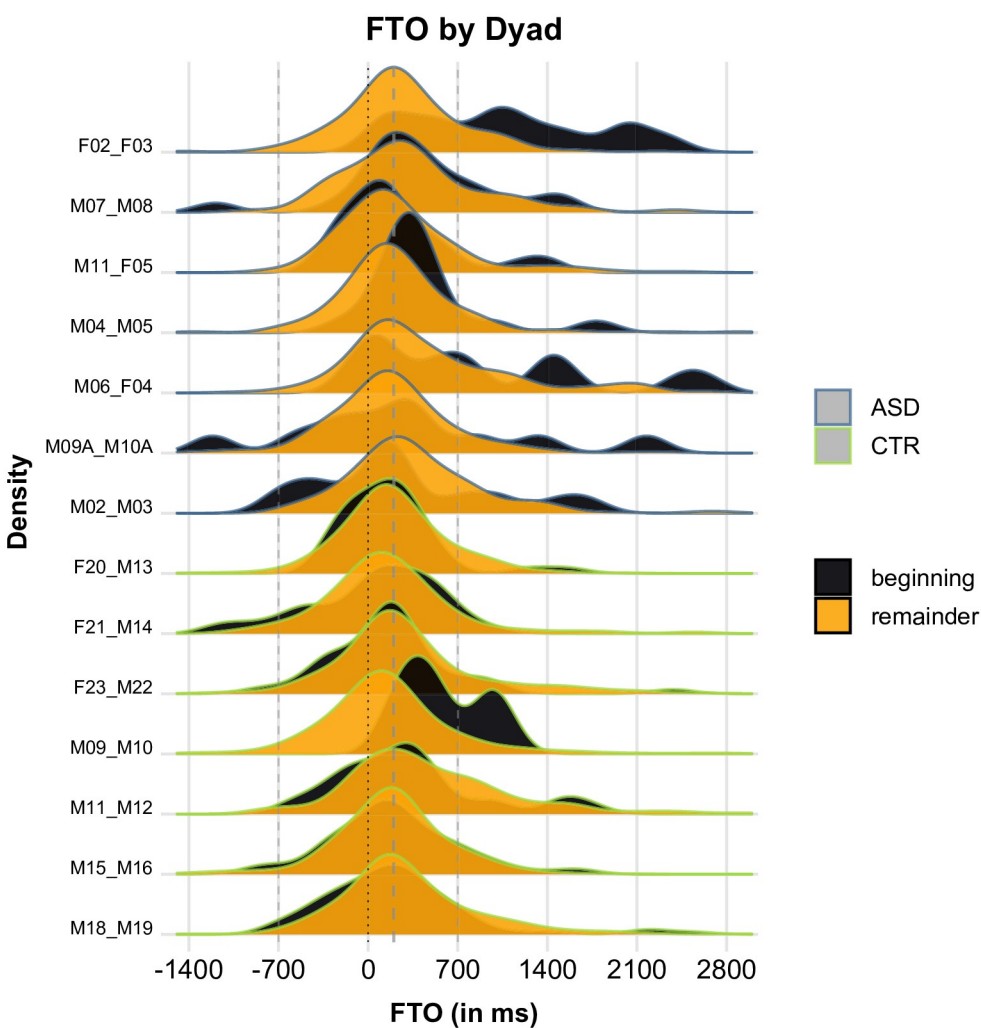

**Fig 7. FTO values by dialogue stage and dyad.** Positive values represent gaps; negative values represent overlaps. ASD dyads in the top half and outlined in blue, CTR dyads in the bottom half and outlined in green. Black curves represent the beginning of dialogue (before detection of the first Mismatch); orange curves represent the remaining dialogue (after detection of the first Mismatch).

shown in related work, through a detailed analysis of all turn transitions directly following the introduction of matching vs. mismatching landmarks, that there was a consistent and distinct reaction to the detection of the first mismatch in both groups (in the form of longer gaps; see [25], Chapter 12.3; [46]). Essentially, the first Mismatch can thus be seen as a turning point in the interaction. Before detection of the first Mismatch, participants might feel that they are expected to give their individual contribution to the solution of a known problem (i.e. draw a path on an otherwise identical map). After the first Mismatch is detected, participants might feel that they need to give a joint contribution to navigate an unknown problem (i.e. the two maps are not identical), and this difference in the conversational goal can be expected to generate a difference in the interaction.

The second reason for using detection rather than resolution is that the former is less problematic as a timestamp from a practical perspective. The time it took to resolve the first Mismatch varied widely across dyads (ranging in duration from under 10 seconds to over 5 minutes; for more detail see [25], Chapter 12). Moreover, even determining when a Mismatch

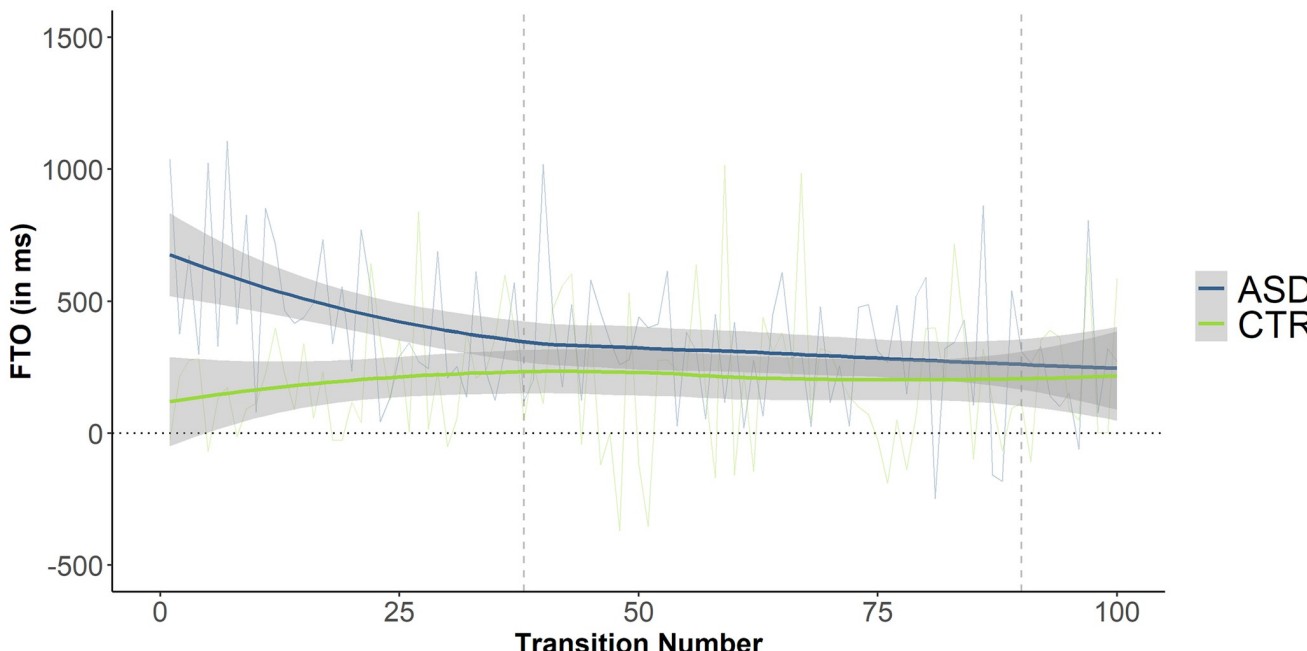

**Fig 8. FTO values by turn transition and group.** Positive values represent gaps; negative values represent overlaps. ASD group in blue, CTR group in green. Thin blue/green lines represent averaged FTO values by transition and group; thick lines represent fitted LOESS-smoothed curves by group, the surrounding grey shaded areas the respective standard error. The dashed vertical lines show 1) transition no. 38 (average time point for detection of first Mismatch) and 2) transition no. 90 (average time point for resolution of first Mismatch).

was in fact "resolved" can be difficult and involves a degree of subjective judgement. In contrast, the detection of the first Mismatch was in almost all cases unambiguously expressed directly in the speech of both interlocutors.

To conclusively examine the appropriateness of using detection of the first Mismatch as the cut-off point, we performed a) a further analysis taking into account the three-way distinction of 1) dialogue from the start of the task to the detection of the first Mismatch, 2) dialogue during the discussion and up to the resolution of the first Mismatch and 3) all remaining (following) dialogue, and b) a continuous analysis of FTO values in the first 100 turn transitions.

Briefly, the analysis with a three-way distinction of dialogue stages confirms that there was a robust between-group difference only before detection, not during and after the discussion of the first Mismatch (details of statistical modelling are reported in the following section).

Finally, Fig 8 shows that average FTO values in the ASD group tended to continuously decrease from the start of conversations until the point when the first mismatch was detected, strengthening the validity of using mismatch detection as a cut-off point. Note that Fig 8 shows only the first 100 turn transitions; dialogues contained a total of 400 transitions on average.

We can conclude that differences in turn-timing were indeed greater between groups in the early stages of dialogue compared to the remainder, independent of the specific cut-off point.

**3.2.2. Statistical analysis.** Bayesian modelling confirms the above description in showing that there was a clear difference in FTO between groups early on in the dialogue, but not at later stages. More details on the interaction between group (CTR vs. ASD) and dialogue stage are given below.

Group differences are reported with the ASD group as the reference level and differences between dialogue stages are reported with the beginning of the dialogue as the reference level.

We first used a Gaussian model with FTO as the dependent variable, the interaction GROUP (ASD/CTR)*DIALOGUE STAGE (BEFORE/AFTER DETECTION OF FIRST MISMATCH) as a fixed factor and DYAD as a random factor. For the comparison of FTO values between groups for only the beginning of the dialogue (i.e., all transitions up to detection of the first Mismatch), the Bayesian model shows a mean $\delta$ of -322 (milliseconds) with a 95% CI of [-462, -183] and a posterior probability $P(\delta > 0) = 1$. The model therefore provides very robust evidence for the observation that autistic dyads produced considerably longer silent gaps between turn transitions than non-autistic dyads in the early stages of dialogue. For the remainder of the dialogue, the mean $\delta$ is -45 (milliseconds) with a 95% CI of [-150, 60] and a posterior probability $P(\delta > 0) = 0.77$. The low posterior probability and the 95% CI including zero by a large margin lend clear support to the observation that there was no difference between the turn-timing of autistic and non-autistic dyads in the later stages of dialogue.

In a three-way distinction of dialogue stages, we can then focus on turn transitions which take place during discussion of the first Mismatch. The relevant model (with the three-way distinction BEFORE/DURING/AFTER DISCUSSION OF FIRST MISMATCH, otherwise identical to the model described directly above) shows that there is no robust group difference for this epoch, expressed through a mean $\delta$ of -98 with a 95% CI [-228, 31] and a posterior probability $P(\delta > 0) = 0.9$. While this indicates a clear trend towards shorter FTO values in non-autistic dyads (in line with the overall trend) during discussion of the first Mismatch, the inclusion of zero in the 95% CI and the relatively low posterior probability suggest that this is not a reliable difference between groups.

## 4. Discussion

In this section, we will summarise our results, situate them in the landscape of previous research, discuss their implications and limitations, and provide a sketch of avenues for future work.

### 4.1. Summary

We presented an in-depth analysis of turn-timing in conversations between dyads of German-speaking adults who either both did or did not have a diagnosis of Autism Spectrum Disorder. We found no overall group-difference in turn-timing. A closer look at different stages of dialogue revealed that autistic dyads did in fact behave differently from control dyads, but only in the earliest stages of dialogue, where they produced more long gaps.

### 4.2. Comparison with previous results

The analysis of conversations between German-speaking adults diagnosed with ASD presented in this work covers unexplored ground in various ways. This rules out a direct comparison of our results with those from previous research for several reasons. First, almost all previous studies on turn-timing in spoken conversations (with one exception) involved autistic children or adolescents (rather than adults, as in ours). Second, most previous research featured spoken interactions between disposition-mixed dyads, that is, between an autistic participant on the one hand and a non-autistic participant on the other (rather than disposition-matched dyads, as in our study). Third, not all previous studies were based on (semi-)spontaneous dialogue (as ours was). Fourth, we are not aware of any directly related work in which participants communicated in German.

With this sizeable caveat in mind, we can most succinctly summarise previous quantitative research on turn-timing in spoken interaction in the context of ASD as reporting a tendency

for more and longer silent gaps in conversations involving autistic individuals. The relevant studies are discussed in more detail in the following paragraphs.

In the first major empirical investigation into turn-taking in autism, [47] report that 12 autistic adolescents and young adults (ages 14–20) produced longer pauses and shorter utterances (and therefore longer gaps) overall than controls, in line with previous, anecdotal observations reported in [48]. However, the generalisability of these results has to be questioned due to three key methodological issues: the age range and intellectual abilities of experimental subjects, the nature of the speech data under consideration, and the methods by which they were elicited (cf. review in [12]). The age range is such that at least some participants have to be assumed to be at different stages of language development, especially as this development tends to be delayed in ASD. Furthermore, no information is given on either general or verbal IQ. Finally, speech data consists of conversations between autistic subjects on one side and either their parents or the experimenters themselves on the other side. Therefore, by the admission of the authors themselves, "the interactions. . .were much more like interviews than unconstrained conversations" [36, p. 453].

More recently, [49] investigated 26 children diagnosed with ASD who were between 4 and 8 years old. All subjects were judged to be verbal and high-functioning. Speech was recorded during administration of the Autism Diagnostic Observation schedule (ADOS [50]), a standardised diagnostic test for ASD. The authors show that autistic children produced longer gaps than age-matched children without a diagnosis of ASD. However, the age (range) of participants and the method of elicitation alone are, each in their own right, reasons enough to preclude reliable conclusions on general strategies of turn-taking in ASD (see also [51] for similar results on Korean).

The authors of [52] investigated day-long naturalistic recordings between children and their parents and found longer silences before responses to question in the ASD compared to a CTR group (see also [53]).

The most recent published work of relevance [54] is notable for featuring adult autistic participants, although they were considerably younger on average than in our sample. Speech data were limited to recordings of the ADOS schedule. Similarly to all the above studies, [54] found a clear tendency for longer silent gaps in the ASD compared to a control group.

Finally, in a meta-analysis of the literature on adult–infant turn-taking, [55] confirm the overall trend for more and longer between-turn silences in conversations involving individuals on the autism spectrum.

One notable departure from this consensus can be found in the wide-ranging and influential "anthropological perspective" put forward in [56]. The authors set out to understand autistic persons not as isolated individuals but rather as social actors with a diverse range of strengths and difficulties in relation to socio-cultural factors and expectations. Crucially, in describing a "cline of competence" across different social domains, [56] report that in the domain of conversational turn-taking, autistic children show few difficulties and "seem to behave qualitatively like many of the unaffected [sic] peers in their families and communities" (p. 162). They speculate that the "local orderliness of sequences" might suit the cognitive style typical for persons on the autism spectrum. Our quantitative findings on autistic adults, revealing no clear overall differences in turn-timing between the ASD and the CTR group, add some support to this earlier qualitative account.

## 4.3. Implications

Our finding that differences in turn-taking between groups, in the form of longer gaps in the conversations of autistic dyads, were found only in the earliest stages of dialogue shows that

autistic speaker pairs successfully established a degree of rapid turn-timing that is essentially indistinguishable from that of non-autistic dyads, but that they did not do so instantly [cf. 57]. Arriving at such equivalent turn-timing behaviour appears to be literally a matter of time for dyads in the ASD group, as it seems to be independent of conversational content (here, progress in the Map Task or, more specifically, encountering and discussing the first Mismatch).

Given that listeners are very sensitive to even small differences in turn-timing [58] and form personality impressions about speakers extremely rapidly [59], the overall turn-taking style of the ASD group may still be perceived as odd or unusual, at least by typically developed listeners, even though there was no robust difference between autistic and non-autistic dyads for most of the dialogue—precisely because the relevant differences are found during the earliest stages of conversation.

These specific differences should, however, not overshadow the general finding that, at a global level, no robust differences were found in conversations between autistic as opposed to non-autistic adults. This might be considered surprising given that a) it has been shown at length in previous work that achieving rapid and precise turn-timing is highly challenging cognitively, as it can only be achieved if speakers are able to accurately predict the communicative intentions of their interlocutor [60–65], and b) predicting the behaviour of others is a skill that many autistic individuals seem to struggle with [66].

Our findings clearly show that at least the kinds of relatively socially motivated and high-functioning autistic adults we investigated, and at least when conversing in disposition-matched (ASD–ASD) dyads, are perfectly able to produce turn-timing of the same speed and precision as has been described for conversations between adults without a diagnosis of Autism Spectrum Disorder. Our observations in related work that speakers in the ASD group did not use more filled pauses than and produced turn-ends and beginnings with the same intonational realisation as the CTR group [25] furthermore discourage alternative explanations for the equivalent turn-timing across the two groups (e.g. that although the utterances of autistic dyads were produced with the same timing, they may have differed in terms of informativeness or prosodic detail). An alternative or complementary theory would be that factors such as Theory of Mind simply play less of a role in turn-taking (and perhaps even ASD in general; see [67]) than has previously been assumed.

Our results extend the numerous findings on the apparent universality of turn-timing for the first time to conversations between autistic adults. On the one hand, this strengthens the notion that turn-taking is a fundamental aspect of human interaction, and one that is apparently very similar across groups of speakers with different cognitive, cultural and linguistic backgrounds. On the other hand, the subtle differences we discovered between the CTR group and the ASD group when taking into account temporal dynamics suggest that similarly subtle differences between other groups of speakers may yet to be discovered. It is possible that a focus on the undeniably remarkable similarities of turn-timing across populations and contexts has overshadowed subtle differences at smaller scales, which might only be discovered with the use of more fine-grained qualitative and quantitative approaches.

### 4.4. Limitations, extensions and future directions

Although we believe that the thoroughness and transparency of our analysis allows us to draw certain conclusions on the basis of the experimental results with a certain degree of confidence, naturally there are many factors to limit external validity.

First, we tested relatively high-functioning, German-speaking autistic adults. There are many ways in which results might differ for individuals situated at different points on the autism spectrum, of different native language backgrounds, and at different stages of

development. The state of the art is such that we cannot directly compare our results to any others on turn-timing in ASD. An obvious extension of the present work would therefore be replications with children and/or with adults speaking a language other than German.

Second, we elicited semi-spontaneous dialogues without eye contact between participants. A multi-modal analysis of video-recorded interactions between speakers with and without ASD could therefore add further crucial information, as gaze and gesture have been shown to play important roles in dialogue management (e.g. [68–73]). Recent work by [74] specifically shows that autistic speakers seemed to use gesture more than non-autistic speakers to regulate turn-timing. We are currently investigating methods of eye-tracking and motion capture for inclusion in future experiments. Regarding the contextual constraints inherent in the Map Task, it is true that having to fulfil an unfamiliar task puts certain pressures and limitations on participants and the resulting linguistic output, and this may have affected speakers in the ASD group differently than those in the CTR group. However, we can speculate that the restricted set of dialogue options and reduced chance of unexpected events may have suited the cognitive styles of autistic speakers more than fully free and spontaneous conversation, which would in turn make the between-group differences described in this paper all the more relevant.

Third, we disregarded prosodic aspects in the present account. While in related work [25] we already conducted an exploratory analysis of the prosodic constructions used in turn-endings and beginnings (among other things) in this data set (following the methodology of [75, 76]), and found no differences between groups, an extended investigation combining quantitative and qualitative methods might prove very fruitful.

Finally, as we investigated the behaviour of disposition-matched dyads (ASD–ASD), perhaps most obvious would be an extension to also include mixed dyads (ASD–CTR). The overwhelming majority of experimental work on communication in ASD has in fact been conducted using mixed dyads only. We chose to record disposition-matched dyads (ASD–ASD) rather than mixed dyads (ASD–CTR) for two main reasons. First, there is quite simply a dramatic lack of research on communication in ASD based on data from such matched dyads. Second, investigating the behaviour of disposition-matched dyads seems to us the most promising way to gain insights into what might justifiably be called "autistic communication". Analysing the behaviour of mixed dyads makes it very difficult to see beyond the patterns and potential difficulties arising from the interaction of individuals with different cognitive styles [77–79]. While such insights are of great value in principle, they cannot be interpreted conclusively and appropriately unless we first have a clear picture of what characterises communication between autistic speakers.

This perspective, in the sense of a certain epistemic humility, extends to the study at hand. For instance, while we can accurately say that the ASD group tended to produce longer silent gaps between turns than the CTR group in certain parts of conversations, by no means can (or should) we claim that this behaviour is simply "wrong" or "inappropriate" in any way. Not only do we have to recognise the very likely possibility that autistic dialogue strategies diverge from those of non-autistic peers in ways that are the most appropriate and functional for this group in the given situation. We also have to acknowledge that we cannot say for sure whether long gaps, produced by any group of speakers, are appropriate or not in a given context without conducting a comprehensive qualitative analysis that takes into account the context of turn transitions. Previous work assures us, for instance, that long gaps are typical and expected in the direct context of verbal exchanges involving misunderstanding or non-alignment [58, 80, 81].

It is beyond the scope of this work to exhaustively analyse how many cases of long gaps were indeed produced in just such contexts for each group, but the detailed analysis of

different stages of dialogue gives us a proxy for such an analysis. We have shown that gaps were longest for the ASD group before detection of the first Mismatch (whereas values were comparable throughout the dialogue for the CTR group). This makes it clear that these cases of long-gap transitions are not specifically linked to unexpected events as part of the task itself, but rather reflect the previously attested observation that people diagnosed with ASD tend to have more difficulties with, and tend to be less comfortable in, situations involving newness or uncertainty. Conversely, we have shown in related work on the same data that dyads from the CTR and the ASD group produced equally long gaps immediately following the introduction of mismatching landmarks, but that dyads from the ASD group produced longer gaps immediately following the introduction of matching landmarks compared to the CTR group ([25], Section 12.3; [46]). In essence, autistic dyads thus produced many long gaps in various situations sharing an element of novelty, while non-autistic dyads only produced long gaps in the context of particularly challenging aspects of the experimental task itself (such as the introduction of an unexpected mismatch). Generally speaking, using such long gaps may be an effective strategy for navigating challenging and unusual situations, and it is employed by both groups of speakers in our data. The difference, then, lies only in the fact that this strategy was used by dyads from the ASD group in a wider variety of contexts.

## 5. Conclusion

We found no clear differences in the global turn-timing of dyads of autistic as compared to non-autistic adults, with all speaker pairs showing a preference for the typically attested short-gap transitions between turns. Differences between groups were instead only found for the early stages of dialogue, in which the ASD dyads clearly differed from CTR dyads in producing longer silent gaps. This finding is of particular relevance because personality judgements and character attributions are disproportionately influenced by the first minutes and even seconds of a social interaction. While ASD dyads took considerably longer to establish a typically rapid exchange of turns, this deviation from the superficially equivalent behaviour between groups at the global level may be functionally motivated, and may indeed represent the most appropriate behaviour for this group of speakers in the early stages of a social interaction. Taking different stages of conversation into account has thus yielded some particularly valuable insights in this first description of turn-timing in conversations between autistic adults.

## Supporting information

**S1 Table. Summary of dialogue duration, number of IPUs and transition types.**
(PDF)

**S1 Fig. Categories of turn transition.** "Gaps" are silent intervals between turn transitions; "between-overlaps" are turn transitions composed of overlapping speech from both interlocutors. "Within-overlaps" are not true floor transfer transitions, but rather represent passages of overlapping speech which are *not* followed by a change of speaker (and therefore did not enter into turn-timing analyses). Adapted from [24].
(TIF)

**S2 Fig. Histograms of FTO values by group.** ASD group in the left panel, CTR group in the right panel. Bin width = 100ms. Positive values represent gaps, negative values represent overlaps.
(TIF)

## Acknowledgments

We would like to thank Martina Krüger for help with recordings as well as the academic editor and two anonymous reviewers for their valuable feedback.

## Author Contributions

**Conceptualization:** Simon Wehrle, Francesco Cangemi, Martine Grice.

**Data curation:** Simon Wehrle, Alicia Janz.

**Formal analysis:** Simon Wehrle.

**Funding acquisition:** Simon Wehrle, Kai Vogeley, Martine Grice.

**Investigation:** Simon Wehrle, Alicia Janz.

**Methodology:** Simon Wehrle, Francesco Cangemi.

**Project administration:** Simon Wehrle, Kai Vogeley, Martine Grice.

**Resources:** Martine Grice.

**Software:** Simon Wehrle, Francesco Cangemi.

**Supervision:** Simon Wehrle, Kai Vogeley, Martine Grice.

**Validation:** Simon Wehrle.

**Visualization:** Simon Wehrle.

**Writing – original draft:** Simon Wehrle.

**Writing – review & editing:** Simon Wehrle, Francesco Cangemi, Alicia Janz, Kai Vogeley, Martine Grice.

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
