## [Decision Letter · Decision Letter 0]

10 Jan 2023

PONE-D-22-26895Turn-timing in conversations between autistic adults: typical short-gap transitions are preferred, but not achieved instantlyPLOS ONE

Dear Dr. Wehrle,

Thank you for submitting your manuscript to PLOS ONE. After careful consideration, we feel that it has merit but does not fully meet PLOS ONE’s publication criteria as it currently stands. Therefore, we invite you to submit a revised version of the manuscript that addresses the points raised during the review process.

Both the two expert reviewers and myself found the research question very interesting, the results clear and the readability of the manuscript good. Still there are some concerns about the description and motivation of the work raised by one reviewer and by myself that should be addressed. According to the reviewer you should better motivate some choices concerning the experimental paradigm (choice of the map task and of the no-eye contact condition) and the analyses (choice Bayesian framework and choice of the first mismatch point). Moreover, some feature of your data should be better described. I would add that section 2.3 should be better organized, that the statistical models used should be explicitly described (dependent variables, predictors and random terms) and that some info is missing in the osf repository (see "Additional Editor Comments" below, where you will also find some textual comment).

We look forward to receiving your revised manuscript.

Kind regards,

Leonardo Lancia, Ph.D

Academic Editor

PLOS ONE

Journal Requirements:

Additional Editor Comments:

The organization of section 2.3 may be improved. In this section you mix the description of the task and of its features with the motivation of its choice (referring to features that have not been introduced yet), and with description the procedure ("...the participants … entered a recording booth..."). I would rather introduce the task and its features, then motivate its choice and finally describe the procedure.

If you refer to statistical analyses, you should provide the means to understand what you did and not only how you did it (which is the content of page 12). Minimally, for each model, you should provide the dependent variable and the fixed and random factors. The reader is referred to the R script used to conduct the analyses that are accessible through an osf repository. However, on the one hand, I don’t feel at ease with the choice of letting the reader look into the scripts and decipher the R code in order to understand the analyses conducted. On the other hand, I think that an important part of the analyses is missing even in the osf repository (the part reported in section 3.2.1). Probably the relevant lines of code are there, but I was not able to find them. Said that, it would be easy to motivate the adoption of a Bayesian approach if the inferences made in section 3.2.1 are based from the posteriors of the model in 3.1.1. For which concerns the markdown file, I think that small paragraphs referring to the manuscript sections would guide the reader more efficiently than the manuscript text itself (but I let you decide what to do in this case).

A final note on the statistical analyses. As the Bayesian approach is the only one used in the manuscript, I would advise to substitute the term “Bayesian” in the sections’ titles with the term “statistical”. As a reader, after reading a section entitled “Bayesian analyses”, I was expecting to find a section with fequentist analyses.

Textual comments

Page 3, lines 50-54. The presence of the labels (a), (b), (c) makes the passage less readable, probably because they do not refer to terms of an enumeration but to sentences which have different functions.

Page 4, lines 68 and 76: evidence of what?

Page 5, line 94: You should give a minimum background for the ICD-10 criteria (e.g. comparable to that provided for the AQ scores).

Page 6, line 117: please define acronyms used (here IQ)

Page 7, line 143: both->the two

Pages 7-10 lines 160-215: Here several topics are mixed, please see general comments.

Page 11, line 255: contains overlap -> overlaps

Page 12, line 273: if the dep variable is FTO, then I would guess that what is tested here is the interaction between group and part of dialogue.

Page 17, line 396: need model specification here and below.

Reviewers' comments:

Reviewer's Responses to Questions

**Comments to the Author**

1. Is the manuscript technically sound, and do the data support the conclusions?

Reviewer #1: Yes

Reviewer #2: Yes

2. Has the statistical analysis been performed appropriately and rigorously? 

Reviewer #1: Yes

Reviewer #2: Yes

3. Have the authors made all data underlying the findings in their manuscript fully available?

Reviewer #1: Yes

Reviewer #2: Yes

4. Is the manuscript presented in an intelligible fashion and written in standard English?

Reviewer #1: Yes

Reviewer #2: Yes

5. Review Comments to the Author

Reviewer #1: This paper presents an analysis of turn-taking in dialogues between pairs of adults with autism spectrum disorders (ASD) compared to pairs of controls. The study finds that globally there is no difference in the turn-taking timing in the different groups, but that there are differences in the initial exchanges, such that the ASD group display greater between turn gaps than the controls. This is a very interesting and timely paper that fills a gap in the literature, which rarely looks at dialogues between people with ASD, instread focussing on dialogues with one control and one ASD individual, and I find the results compelling and very interesting. However, before the paper can be published, there needs to be some more work done on motivating some of the choices made, as well as some acknowledgement of how such decisions might affect the interpretation of the results.

In the Introduction, the authors state (l.63) that "successful and rapid turn-taking crucially relies on socio-communicative abilities related to pragmatics and perspective-taking". In fact, this is not universally accepted to be the case, and the authors need to cite some relevant background material if they wish to make this claim.

Motivations for the use of the Map Task should be stronger -- the Map Task, while well studied, is of course a particular genre of dialogue which may not resemble more spontaneous convrsations. This is fine, since the authors can argue that while this is true, both groups have the same task, but the nature of the task might plausibly be more difficult for the ASD group, so this at least needs acknowledgement. Similarly, there should be some motivation for using the no eye contact version of the task, since this is known to produce different patterns of dialogue than when eye-contact is permitted (particularly in terms of backchannels and verbal repairs, which are relevant for the measures in this study). Once again, I don't think this is problematic since this is one of the first studies on dialogue between ASD adults, but it needs to be motivated. More explanation of the data (l.223) is also necessary -- for example, I assume that the reason there are fewer transitions than IPUs is because the IPUs include within-speaker pauses, but this needs to be explicit.

I'm puzzled by the 'within-overlap' cases (l.250). This needs to be more clearly explained. And if 70% of cases were typically very short backchannels, what were the other 30%? It would also be useful to know how these cases were identified. In fact, it would be good to have a table of the values detailed in the paragrah from l.259 by groups as well as in total, as well as some headline figures to see how equivalent the dialogues were (total number of words/turns, durationetc).

The use of Bayesian modelling needs to be motivated -- why did you use these techniques (they're not wrong, but will not be as familiar to most readers that null hypothesis testing techniques, so there needs to be an explanation of why they are better for this task).

It was never clear to me why the first mismatch was chosen as the relevant point in the dialogue for the analysis. This really needs to be properly motivated. Why was this chosen? What are the characteristics of this point in the dialogue in the different groups (how long are the different segments in the dialogues etc?). This seems like it might be an overly complex measure, and really needs a lot of work to justify (what is it about the first mismatch identification which marks the transition from the early part of the dialogue to the rest -- this seems arbitrary). It may be that the authors have valid reasons for choosing this point, but it was not obvious to me. There may be more 'stupid' measures that also show the pattern of results and don't require the same burden of explanation (e.g. the first 30 seconds of the dialogues). I'm especially concerned about this measure since the authors rely so heavily on it, but also state that there was a large range in values (particularly when they consider the mismatch resolution point, l.420). This really begs the question to me: Why use the first mismatch at all? This also seems counter to the statements of the authors that they are not interested in the spoken parts of the dialogue, since this is very context and content dependent.

Para at l.552 needs references (at least point a) is not universally accepted, as indeed the authors seem to entertain in l.572). The assertion at l.565 "although it is likely that this process is still relatively more effortful for autistic individuals" seems to come from nowhere. Is it? Why? l.642 needs citations -- what previous work? -- At least the work I am aware of on repair suggests that this usually occurs extremely locally and rapidly -- more extensive misunderstandings (that only get noticed further downstream) may result in long gaps but this does not seem to be generally true.

Reviewer #2: 1. Is the manuscript technically sound, and do the data support the conclusions?

The manuscript is very technically sound and I think the conclusions they draw are very clear and very well-supported by the data. It would have been quite tempting to speculate beyond what the data seem to say (that differences between experimental groups exist in only one region of the dialogues, namely the beginnings, and in only one way), but the authors refrain. I think the paper is very tight, and I think the simplicity of the questions and analysis result in the manuscript's contribution being very clear and compelling.

2. Has the statistical analysis been performed appropriately and rigorously?

I think the Bayesian analyses of the patterns (which were actually pretty clear-cut) were appropriate. These are tools that, in our field/subfield are very current and have some advantages in terms of rigor over some alternative methods that are also currently used in our field/subfield.

3. Have the authors made all data underlying the findings in their manuscript fully available?

As I understood it, the authors have made the data fully available.

4. Is the manuscript presented in an intelligible fashion and written in standard English?

The paper is written in standard English, and indeed very clear prose -- one of the paper's strengths.

Final Comments:

I think this paper makes a very important and clear contribution to the literature on turn-taking, in both special and typical populations. In brief, the authors have brought an unusual level of clarity and rigor to this topic, and they have framed their study in a way that I believe will influence future work in a positive way. I also think the limitations and caveats that they authors highlight are the right ones. All of this said, and given the paper's overall brevity and concision, I do not see a reason to not accept this paper -- I myself have no requests to make in a revision.

6. PLOS authors have the option to publish the peer review history of their article (what does this mean?). If published, this will include your full peer review and any attached files.

Reviewer #1: No

Reviewer #2: No

---

## [Author Response · Author response to Decision Letter 0]

18 Feb 2023

!!! Please ONLY refer to the pdf version of the Response to Reviewers for proper formatting !!!

(pasted here only as this was strictly required for proceeding with the re-submission)

PONE-D-22-26895

Turn-timing in conversations between autistic adults: typical short-gap transitions are preferred, but not achieved instantly

PLOS ONE

Response to Reviewers

We would like to thank both reviewers as well as the editor sincerely for the time and effort they invested into considering our submission. We believe their comments have helped to strengthen the paper in its new, revised form. We respond to all comments and questions point-by-point below (each starting on a new page, except for the textual comments by the editor). Comments by the editor and the reviewers are copied in italics and smaller font, our responses are interspersed in regular type and excerpts from the revised manuscript (with tracked changes, and the line numbers from the respective document) are indented. Please let us know if there are any further questions. We thank you for your further consideration of this article and hope you will be satisfied with the revisions we implemented.

Kind regards

Simon Wehrle, Cologne, 4 February 2023

(also on behalf of my co-authors)

Editor’s comments

Both the two expert reviewers and myself found the research question very interesting, the results clear and the readability of the manuscript good. Still there are some concerns about the description and motivation of the work raised by one reviewer and by myself that should be addressed. According to the reviewer you should better motivate some choices concerning the experimental paradigm (choice of the map task and of the no-eye contact condition) and the analyses (choice Bayesian framework and choice of the first mismatch point). Moreover, some feature of your data should be better described. I would add that section 2.3 should be better organized, that the statistical models used should be explicitly described (dependent variables, predictors and random terms) and that some info is missing in the osf repository (see "Additional Editor Comments" below, where you will also find some textual comment).

Thank you very much for this assessment and for your time. We have addressed all further comments and concerns, as detailed below.

The organization of section 2.3 may be improved. In this section you mix the description of the task and of its features with the motivation of its choice (referring to features that have not been introduced yet), and with description the procedure ("...the participants … entered a recording booth..."). I would rather introduce the task and its features, then motivate its choice and finally describe the procedure.

Thank you for this very useful comment, which I believe has helped us to improve the clarity of the final paper. Section 2.2 (which I believe the editor is referring to, rather than 2.3) has been rearranged and rewritten, with a now far clearer separation of task description, procedure and motivation of the choice (we used this order for better flow of the overall manuscript). This also includes some added justifications for choosing the Map Task paradigm, in response to Reviewer 1 (see their comment below). Copied from line 181:

2.2 Materials and procedure

We used Map Tasks to elicit semi-spontaneous speech. The Map Task paradigm was introduced by [16] and has widely been used in speech research for over 30 years (see [17] for an influential article describing a corpus of Map Task speech). We chose this paradigm as it provides us with predominantly spontaneous speech data that can, however, still be controlled along a number of key parameters, such as lexical items (via the names of landmarks on a map) and communicative obstacles (such as the introduction of mismatching landmarks between maps; see below for more detailThe Map Task paradigm was introduced by [18] and has widely been used in speech research for over 30 years (see [19] for an influential article describing a corpus of Map Task speech).

After filling in a number of forms and the questionnaires listed in Section 2.1, participants received written instructions for the task and entered a recording booth. Each participant was presented with a simple map containingMaterials consisted of pairs of simple maps. Each map contained 9 landmark items in the form of small pictures (materials adapted from [18]). Only one of the two participants[20]). Only one of the two participants in each Map Task (the instruction giver) had a route printed on their map. The experimental task was for the instruction follower to transfer this route to their own map by exchanging information with the instruction giver. During this entire process, an opaque screen was placed between participants, meaning they could not establish visual contact and had to solve the task by means of verbal communication alone. The roles of instruction giver and instruction follower were assigned randomly. Upon completion of the first task, subjects received a new set of maps and their roles were switched. The task ended once the second Map Task was completed. 

As participants were naive to the purpose of the study, they did not know at the outset that their maps differed in some crucial regards. In each map, some landmarks were either missing, duplicated and/or replaced with a different landmark, compared to the interlocutor’s map. This was the case for 2 landmarks per map. Those items that differed between maps will hereafter be called Mismatches; items that were the same on both maps will be called Matches. During annotation, we marked the portion of dialogue in which the first Mismatch was discussed by participants and used it to divide all dialogues up into three epochs, i.e., before detection, during discussion, and after resolution of the first Mismatch (see Section 3.2 for further details). An example of maps used in this study is shown in Fig 1, with Mismatches highlighted using red circles. All dyads received the same two pairs of maps.

Map Task

Fig 1. Example of Map Task materials. The instruction giver’s map, with a route leading from “Start” (top left) to “Ziel” (finish; bottom left), is in the left panel. Mismatches between maps are highlighted with red circles.

Participants in the study first filled in a number of forms and the questionnaires listed in Section 2.1, then received written instructions for the task and finally entered a recording booth. They then received one map each (only one of which featured the route from start to finish). During this entire process, an opaque screen was placed between participants, meaning they could not establish visual contact and had to solve the task by means of verbal communication alone. We chose to restrict conversations (and the subsequent analysis) to the spoken modality as we were not equipped to perform in-depth analyses of multi-modal interaction at the time of recording. The roles of instruction giver and instruction follower were assigned randomly. Upon completion of the first task, subjects received a new set of maps and their roles were switched. The task ended once the second Map Task was completed. As participants were naive to the purpose of the study, they did not know at the outset that their maps differed in some crucial regards.

All conversations were recorded in a sound-proof booth at the Department of Phonetics, University of Cologne. We used two head-mounted microphones (AKG C420L) connected through an audio-interface (PreSonus AudioBox 22VSL) to a PC running Adobe Audition. The sample rate was 44100 Hz (16 bit). Recordings were transcribed orthographically and divided into interpausal units (IPUs) with a minimum pause length of 200 ms.

Fig 1. Example of Map Task materials. The instruction giver’s map, with a route leading from “Start” (top left) to “Ziel” (finish; bottom left), is in the left panel. Mismatches between maps are highlighted with red circles.

We only included recorded dialogue from the start to the end of each task in all analyses, in order to achieve a greater degree of comparability regarding conversational context and content. The total duration of speech material is 4 hours and 44 minutes. The mean dialogue duration is 20 minutes and 19 seconds (SD = 12’32’’). 

Figure 2 shows an example excerpt of Map Task dialogue from one of the ASD dyads, transcribed following GAT conventions [19,20].[21,22]. Two examples of turn transitions are highlighted in bold, one following the introduction of a matching landmark — “heller Diamant”, line 15/16—and one following the introduction of a mismatching landmark—“goldene Moschee”, line 21/22. Note that the turn transitions highlighted here are considerably longer than average transitions between turns.

Fig 2. Example excerpt of a GAT transcription. Two turn transitions (following newly introduced landmarks) are highlighted in bold (lines 15/16; 21/22).

We chose the Map Task paradigm for the current investigation as it provides us with predominantly spontaneous speech data that can, however, still be controlled along a number of key parameters, such as lexical items (via the names of landmarks on a map) and communicative obstacles (such as the introduction of mismatching landmarks between maps). While the elicited dialogues are not fully free or spontaneous, the Map Task seemed to us a good choice in the context of comparing autistic and non-autistic dyads, since the constraints involved in the task serve to reduce a potentially particularly high degree of variability across the autism spectrum in terms of social motivation, interest in a given topic, and the adherence to social conventions.

If you refer to statistical analyses, you should provide the means to understand what you did and not only how you did it (which is the content of page 12). Minimally, for each model, you should provide the dependent variable and the fixed and random factors.

Thank you for this remark, which will help clarify analyses for our readers. For all sections reporting on statistical analyses (2.1, 3.1.1, 3.2.2.), we have now added information on the dependent variables as well as fixed and random factors. To wit (copied from line 160, 433 and 560, respectively):

Most importantly, there was a clear difference in AQ scores between groups, with a far higher average score in the ASD group (mean = 41.9; range = 35–46) than in the CTR group (mean = 16.1; range: 11–26) and no overlap at all between subjects from boththe two groups. Bayesian modelling provides unambiguous evidence for the group difference in AQ scores, and also confirms that the differences in age and verbal IQ are small but robust. Table 1 shows summary statistics for gender, age, verbal IQ and AQ. (We used Poisson models with Age/Verbal IQ/AQ as the respective dependent variable, and Group (ASD/CTR) as the independent variable in all cases.)

(…)

3.1.1 BayesianStatistical analysis

All modelsWe used for Bayesian analysis includeda Gaussian model with FTO as the dependent variable, Group (ASD/CTR) as a fixed factor and Dyad as a random intercepts for dyads (full specifications in the accompanying repository at https://osf.io/v5pn4/).factor. The group difference in the model is reported with the ASD group as the reference level

(…)

Group differences are reported with the ASD group as the reference level and differences between dialogue stages are reported with the beginning of the dialogue as the reference level. In aWe first used a Gaussian model with FTO as the dependent variable, the interaction Group (ASD/CTR)*Dialogue Stage (before/after detection of first mismatch) as a fixed factor and Dyad as a random factor. (…) It thus only remains to explicitly test turn transitions which take place during discussion of the first Mismatch. The relevant modelThe relevant model (with the three-way distinction Before/During/After Discussion of First Mismatch, otherwise identical to the model described directly above) shows that there is no robust group difference for this epoch, expressed through a mean 𝛿 of -98 with a 95% CI [-228, 31] and a posterior probability 𝑃 (𝛿 > 0) = 0.9.

The reader is referred to the R script used to conduct the analyses that are accessible through an osf repository. However, on the one hand, I don’t feel at ease with the choice of letting the reader look into the scripts and decipher the R code in order to understand the analyses conducted. On the other hand, I think that an important part of the analyses is missing even in the osf repository (the part reported in section 3.2.1). Probably the relevant lines of code are there, but I was not able to find them. 

Thank you for this remark, and for being tolerant of our perhaps somewhat unorthodox approach, which we hope is also a progressive one. The main intention is to provide full transparency in making available every line of code and data, but at the same time also greatly improving the readability and flow of the manuscript itself by not directly reporting all details on statistical analysis in all cases.

We apologise if the script/markdown file remained opaque for the analysis you mention, despite our best efforts. The relevant parts of the code you refer to as having missed can now be found in lines 1149–1468 of the markdown file. We restructured the file in an effort to improve clarity and readability. 

Said that, it would be easy to motivate the adoption of a Bayesian approach if the inferences made in section 3.2.1 are based from the posteriors of the model in 3.1.1. For which concerns the markdown file, I think that small paragraphs referring to the manuscript sections would guide the reader more efficiently than the manuscript text itself (but I let you decide what to do in this case).

We are grateful for this well-considered remark. However, we believe it would not be appropriate in this case to use the outcomes from one analysis to directly inform the choice and specification of priors for another analysis from the same data set (although this could be considered for future studies with new data). We instead deliberately chose to employ weakly informative priors for all analyses (see references [31,33,40]).

As for motivating the adoption of a Bayesian approach, we added some comments pointing to what we perceive to be some of the most relevant advantages of a Bayesian approach in Section 2.3 (see below, copied from line 341). Thank you for leaving us with the final decision on what to include in the Rmd files. As we intend the Rmd file to function as stand-alone document, we have decided to keep (and update) all text from the manuscript in place, rather than only using pointers to sections/chapters. We hope you understand this decision.

In reporting the results of this essentially exploratory study, we emphasise detailed description and data visualisation [26,27] along with an in-depth analysis of dyad-specific behaviour [28–31]. We use Bayesian inference to corroborate our findings, but consider descriptive, exploratory analysis to be at the heart of this work. Therefore, we report the most essential elements of the Bayesian models used in the paper itself, but for all further details, as well as all data frames, scripts and codes used to generate the analyses and plots in this paper, we refer the reader to the accompanying OSF repository at https://osf.io/v5pn4.

We chose to use Bayesian rather than frequentist statistics for a number of reasons. First, given the limited sample size of the study at hand as well as the scant previous research on the topic, we deem presenting our results and analysis as exploratory, rather than confirmatory, as the best option. Bayesian inference is particularly well suited to studies with a limited sample size, as this limitation can be directly reflected in the model output (e.g. in the form of larger credible intervals and a lower posterior probability). The method gives outcomes based on the data at hand, the chosen model and the specified prior assumptions. Compared to frequentist inference, it is therefore, when properly applied, more conservative, but also more robust and transparent than frequentist approaches [32–35]. Second, Bayesian inference is rapidly increasing in popularity in linguistics and many other fields. This is due in part to practical reasons, as recent statistical software, tutorials and packages have made the application of Bayesian multilevel modelling increasingly straightforward and at the same time considerably more robust and flexible than the frequentist alternatives [36]. Additionally, Bayesian methods seem to be much more closely aligned with common human intuitions and ways of reasoning about the interpretation of statistical tests in general and the notion of significance in particular [35,37,38].

A final note on the statistical analyses. As the Bayesian approach is the only one used in the manuscript, I would advise to substitute the term “Bayesian” in the sections’ titles with the term “statistical”. As a reader, after reading a section entitled “Bayesian analyses”, I was expecting to find a section with fequentist analyses.

Thank you for spotting these potentially misleading section headings. We replaced headings for 3.1.1. and 3.2.2. (formerly 3.2.1; we have now added a new section 3.2.1. dedicated to exploring different cut-offs (including a new one) for “early” vs. “late” dialogue stages in response to Reviewer 1) with “Statistical analysis”.

Textual comments

Page 3, lines 50-54. The presence of the labels (a), (b), (c) makes the passage less readable, probably because they do not refer to terms of an enumeration but to sentences which have different functions.

Thank you for spotting this. We hope the new formulation is more readable (copied from line 49):

Human turn-taking in conversation, however, is a unique and remarkable phenomenon, however, because (a) it is. It is not only executed with split-second precision and flexibilityy,, (b) it involves as well as and involvinges the parallel prediction, planning and production of utterances which are improvised yet rich with meaning and (c), but it is also the key means through which human language, and to a considerable extent human culture, are learned and transmitted (cf. [3]).

Page 4, lines 68 and 76: evidence of what?

Very good point. We believe this is a matter of imprecise wording more than anything else—“evidence” has been replaced with “research” and “analysis”, respectively (copied from line 73):

However, there is only scantvery limited quantitative evidenceresearch on turn-timing in Autism Spectrum Disorder (ASD) to date, and none whatsoever on turn-timing in conversations between autistic adults to date. The limited experimental evidence available seems to point to a general tendency for longer silent gaps in conversations involving autistic participants (see Section 4.2), although it is not clear to which extent this trend can be expected to apply to (semi-)spontaneous conversations between autistic adults, which are investigated in this study for the first time. 

We present experimental evidence on strategiesan analysis of turn-taking strategies in pairs of German adults, where both interlocutors either did or did not have diagnosis of ASD (in so-called disposition-matched dyads). When considering the dialogue as a whole, we found no clear differences in turn-timing between the ASD and the control (CTR) group. However, closer inspection reveals that, compared to control speakersCTR dyads, autistic dyads produced longer gaps between turns specifically in the earliest stages of dialogue. We discuss the implications of these results and relate them to previous research on autism and to the notion of seemingly universal patterns of turn-timing in spoken dialogue.

Page 5, line 94: You should give a minimum background for the ICD-10 criteria (e.g. comparable to that provided for the AQ scores).

We added the key diagnostic criteria (copied from line 98):

Participants from the ASD group had all been diagnosed with autism (corresponding to ICD-10: F84.0; see [9][11]) or Asperger syndrome (ICD-10: F84.5) and were recruited in the Autism Outpatient Clinic at the Department of Psychiatry of the University of Cologne (Germany). The key diagnostic criteria described in the ICD-10 are 1) unusual (“impaired”) social interaction and communication and 2) a restricted repertoire of activities and interests. 

Page 6, line 117: please define acronyms used (here IQ)

The definition was added (copied from line 133):

Although participants from the CTR group were matched as closely as possible to the ASD group for age, verbal IQ (intelligence quotient) and gender, some minor differences remained.

Page 7, line 143: both->the two

Changed accordingly (copied from line 160):

Most importantly, there was a clear difference in AQ scores between groups, with a far higher average score in the ASD group (mean = 41.9; range = 35–46) than in the CTR group (mean = 16.1; range: 11–26) and no overlap at all between subjects from boththe two groups.

Pages 7-10 lines 160-215: Here several topics are mixed, please see general comments.

The whole section was rewritten, as detailed in the response to the comment above.

Page 11, line 255: contains overlap -> overlaps

Thank you for spotting the missing article, we further changed it to “a period of overlap” to also account for comments from Reviewer 1 (see below) regarding the within-overlap category (pasted from line 312):

The related category of “within-overlaps”, ” refers to cases in which part of one speaker’s ongoing turn contains a period of overlap with speech from the interlocutor (, but is not followed by a change of speaker) does not in fact [24].

Page 12, line 273: if the dep variable is FTO, then I would guess that what is tested here is the interaction between group and part of dialogue.

That’s correct and the text has been amended accordingly; thank you for noticing this oversight (copied from line 368):

For the main analysis, we tested for group differences in FTO values as well as the interaction of FTO valuesgroup with part of dialogue (beginning vs. remainder; see Section 3.2 for details).

Page 17, line 396: need model specification here and below.

Model specifications were added as per your comment (and our response above, see copied sections there), i.e. dependent variable, fixed and random factors are now listed for each model.

Reviewer 1

This paper presents an analysis of turn-taking in dialogues between pairs of adults with autism spectrum disorders (ASD) compared to pairs of controls. The study finds that globally there is no difference in the turn-taking timing in the different groups, but that there are differences in the initial exchanges, such that the ASD group display greater between turn gaps than the controls. This is a very interesting and timely paper that fills a gap in the literature, which rarely looks at dialogues between people with ASD, instread focussing on dialogues with one control and one ASD individual, and I find the results compelling and very interesting. However, before the paper can be published, there needs to be some more work done on motivating some of the choices made, as well as some acknowledgement of how such decisions might affect the interpretation of the results.

We are very grateful for your review and the thoughtful and constructive feedback.

We respond to all individual comments in turn below.

In the Introduction, the authors state (l.63) that "successful and rapid turn-taking crucially relies on socio-communicative abilities related to pragmatics and perspective-taking". In fact, this is not universally accepted to be the case, and the authors need to cite some relevant background material if they wish to make this claim.

Thank you, and we agree that including perspective-taking in this sentence was an unjustified overstatement. We now refer simply to pragmatic language skills, and also added some general references for this assertion (copied from line 68):

As successful and rapid turn-timing crucially relies on socio-communicative abilities such as pragmatic language skills [4,9,10], which are typically thought to be impaired in individuals on the autism spectrum, delayed or otherwise divergent patterns of turn-timing in this population might plausibly be predicted.

Motivations for the use of the Map Task should be stronger -- the Map Task, while well studied, is of course a particular genre of dialogue which may not resemble more spontaneous convrsations. This is fine, since the authors can argue that while this is true, both groups have the same task, but the nature of the task might plausibly be more difficult for the ASD group, so this at least needs acknowledgement. Similarly, there should be some motivation for using the no eye contact version of the task, since this is known to produce different patterns of dialogue than when eye-contact is permitted (particularly in terms of backchannels and verbal repairs, which are relevant for the measures in this study). Once again, I don't think this is problematic since this is one of the first studies on dialogue between ASD adults, but it needs to be motivated. 

Thank you for pointing this out. We agree there should be clearer motivation for choosing the Map Task in this particular context, and that limitations could be stated more explicitly (although we have a different intuition as to the specific difficulty of this scenario for the ASD group, as cautiously stated below). In addition to the relevant sections that were present in the original manuscript (in the limitations section, line 793+ in the revised manuscript with tracked changes), we have added the following (copied from lines 229, 267 and 793, respectively). :

During this entire process, an opaque screen was placed between participants, meaning they could not establish visual contact and had to solve the task by means of verbal communication alone. We chose to restrict conversations (and the subsequent analysis) to the spoken modality as we were not equipped to perform in-depth analyses of multi-modal interaction at the time of recording.

(…)

We chose the Map Task paradigm for the current investigation as it provides us with predominantly spontaneous speech data that can, however, still be controlled along a number of key parameters, such as lexical items (via the names of landmarks on a map) and communicative obstacles (such as the introduction of mismatching landmarks between maps). While the elicited dialogues are not fully free or spontaneous, the Map Task seemed to us a good choice in the context of comparing autistic and non-autistic dyads, since the constraints involved in the task serve to reduce a potentially particularly high degree of variability across the autism spectrum in terms of social motivation, interest in a given topic, and the adherence to social conventions.

(…)

Second, we elicited semi-spontaneous dialogues without eye contact between participants. A multi-modal analysis of video-recorded interactions between speakers with and without ASD could therefore add further crucial information. As our analysis was designed to focus on spoken language only, we blocked other modalities in the experimental set-up by installing an opaque screen between interlocutors. However, gaze and gesture also play important roles in dialogue management, as discussed in e.g. [51–56]. Recent work by [57] specifically showed that autistic speakers seemed to use gesture more than non-autistic speakers to regulate turn-timing. We are currently investigating methods of eye-tracking and motion capture for inclusion in future experiments. , as gaze and gesture have been shown to play important roles in dialogue management (e.g. [68–73]). Recent work by [74] specifically shows that autistic speakers seemed to use gesture more than non-autistic speakers to regulate turn-timing. We are currently investigating methods of eye-tracking and motion capture for inclusion in future experiments. Regarding the contextual constraints inherent in the Map Task, it is true that having to fulfil an unfamiliar task puts certain pressures and limitations on participants and the resulting linguistic output, and this may have affected speakers in the ASD group differently than those in the CTR group. However, we can speculate that the restricted set of dialogue options and reduced chance of unexpected events may have suited the cognitive styles of autistic speakers more than fully free and spontaneous conversation, which would in turnfacilitating make between-group comparisons and potentially making the between-group differences foundhave described in this paper here all the more relevant.

More explanation of the data (l.223) is also necessary -- for example, I assume that the reason there are fewer transitions than IPUs is because the IPUs include within-speaker pauses, but this needs to be explicit.

We apologise for the lack of clarity in the description of the data. A new sentence was added to address this (copied from line 280):

Our data set contains 18332 IPUs in total (inter-pausal units; here defined as speech separated by at least 200 milliseconds of silence). For an analysis of turn-taking, not thethese units of speech in themselves are of primary interest, but rather the points of transition between them. Our data set contains 5668 such transitions overall. There are fewer turn transitions than IPUs because most of the latter were followed by another IPU from the same speaker; i.e. separated by within-speaker pauses rather than between-speaker gaps.

I'm puzzled by the 'within-overlap' cases (l.250). This needs to be more clearly explained. And if 70% of cases were typically very short backchannels, what were the other 30%? It would also be useful to know how these cases were identified. In fact, it would be good to have a table of the values detailed in the paragrah from l.259 by groups as well as in total, as well as some headline figures to see how equivalent the dialogues were (total number of words/turns, durationetc).

Thank you for pointing out this lack of clarity. We made our best attempt to elucidate this further with these additions (additional text, plus one supplementary figure and one supplementary table, from line 312 plus Supporting Information):

The related category of “within-overlaps”, ” refers to cases in which part of one speaker’s ongoing turn contains a period of overlap with speech from the interlocutor (, but is not followed by a change of speaker) does not in fact [24]. In other words, these are situations where Speaker A has started and continues speaking, Speaker B then produces a simultaneous utterance (e.g. “yes”), but then falls silent again, with only Speaker A continuing to speak (see S1 Fig,; Supporting Information). This does not entail a floor transfer from one speaker to another and such cases did therefore not enter into the analysis of turn-timing that is the main focus of this paper. Briefly, distribution and characteristics of within-overlaps were very similar in ASD and CTR conversations: they were typically very short (around 380 ms) and contained a backchannel (a backchannel tokensling (listener signalsignals such as “mmhm” or “yeah”) in around 70% of cases, for both groups. (and e.g. answers to tag questions, or longer utterances, in the other cases; see e.g. Part IV of [25], Part IV, for further details on the analysis of backchannels).

Of the 5668 transitions in the data set, 3418 were silent gaps (60.3%), 1326 were between-overlaps (23.3%) and 924 were within-overlaps (16.3%). After the exclusion of within-overlaps, 4744 transitions remained for the analysis of turn-timing. 72% of these were gaps, 28% (between-)overlaps. (further information in S1 Table; Supporting Information).

(…)

S1 Table. Summary of dialogue duration, number of IPUs and transition types.

Group

Mean dialogue duration (SD)

Total IPUs

Total turn transitions

Total silent gaps (% of all transitions)

Total between-overlaps (% of all transitions)

Total within-overlaps (% of all transitions)

ASD

14’ 37’’ 

(7’ 12’’) 

6211

1841

1168 

(63.4 %)

388

(21.1 %)

285 

(15.5 %)

CTR

26’ 01’’ 

(14’ 35’’) 

12121

3827

2250 

(58.8 %)

938 

(24.5 %)

639 

(16.7 %)

Total

20’ 19’’

(12’ 32’’)

18332

5668

3418 (60.3 %)

1326 (23.4 %)

924 (16.3 %)

S1 Fig. Categories of turn transition. “Gaps” are silent intervals between turn transitions; “between-overlaps” are turn transitions composed of overlapping speech from both interlocutors. “Within-overlaps” are not true floor transfer transitions, but rather represent passages of overlapping speech which are not followed by a change of speaker (and therefore did not enter into turn-timing analyses). Adapted from [24].

The use of Bayesian modelling needs to be motivated -- why did you use these techniques (they're not wrong, but will not be as familiar to most readers that null hypothesis testing techniques, so there needs to be an explanation of why they are better for this task).

We appreciate this remark and hope an explicit explanation for using Bayesian modelling will strengthen the paper. Please see the above response to a similar comment by the editor (emphasising the need for further information here), copied for convenience:

As for motivating the adoption of a Bayesian approach we added some lines pointing to what we perceive to be some of the most relevant advantages of a Bayesian approach in Section 2.3 (copied from line 341):

In reporting the results of this essentially exploratory study, we emphasise detailed description and data visualisation [26,27] along with an in-depth analysis of dyad-specific behaviour [28–31]. We use Bayesian inference to corroborate our findings, but consider descriptive, exploratory analysis to be at the heart of this work. Therefore, we report the most essential elements of the Bayesian models used in the paper itself, but for all further details, as well as all data frames, scripts and codes used to generate the analyses and plots in this paper, we refer the reader to the accompanying OSF repository at https://osf.io/v5pn4.

We chose to use Bayesian rather than frequentist statistics for a number of reasons. First, given the limited sample size of the study at hand as well as the scant previous research on the topic, we deem presenting our results and analysis as exploratory, rather than confirmatory, as the best option. Bayesian inference is particularly well suited to studies with a limited sample size, as this limitation can be directly reflected in the model output (e.g. in the form of larger credible intervals and a lower posterior probability). The method gives outcomes based on the data at hand, the chosen model and the specified prior assumptions. Compared to frequentist inference, it is therefore, when properly applied, more conservative, but also more robust and transparent than frequentist approaches [32–35]. Second, Bayesian inference is rapidly increasing in popularity in linguistics and many other fields. This is due in part to practical reasons, as recent statistical software, tutorials and packages have made the application of Bayesian multilevel modelling increasingly straightforward and at the same time considerably more robust and flexible than the frequentist alternatives [36]. Additionally, Bayesian methods seem to be much more closely aligned with common human intuitions and ways of reasoning about the interpretation of statistical tests in general and the notion of significance in particular [35,37,38].

It was never clear to me why the first mismatch was chosen as the relevant point in the dialogue for the analysis. This really needs to be properly motivated. Why was this chosen? What are the characteristics of this point in the dialogue in the different groups (how long are the different segments in the dialogues etc?). This seems like it might be an overly complex measure, and really needs a lot of work to justify (what is it about the first mismatch identification which marks the transition from the early part of the dialogue to the rest -- this seems arbitrary). It may be that the authors have valid reasons for choosing this point, but it was not obvious to me. There may be more 'stupid' measures that also show the pattern of results and don't require the same burden of explanation (e.g. the first 30 seconds of the dialogues). I'm especially concerned about this measure since the authors rely so heavily on it, but also state that there was a large range in values (particularly when they consider the mismatch resolution point, l.420). This really begs the question to me: Why use the first mismatch at all? This also seems counter to the statements of the authors that they are not interested in the spoken parts of the dialogue, since this is very context and content dependent.

Thank you for raising this important issue. We are grateful for this observation and think that the additions we made to address it serve to greatly strengthen the paper as a whole. We hope you will agree. We added a new section, 3.2.1: Corroboration of dialogue stage effect, to fully explain the choice and relevance of the first Mismatch as a cut-off point. We additionally present a new Fig 8 with continuous FTO data from the first 100 turn transitions, thus removing the need for choosing and motivating any one specific “blind” (/”stupid”) threshold in the paper. For your information, before viewing the continuous analysis, we decided to choose 10% of the average number of turn transitions per dialogue (400) as a blind cut-off for the early stage of dialogue, and this point (transition no. 40) happens to coincide almost perfectly with the average time point when the first mismatch was detected (transition no. 38, see Fig 8). Copied from line 491:

3.2.1 Corroboration of dialogue stage effect

In all the above analyses, we used detection (i.e. first mention) rather than resolution of the first Mismatch (i.e. the time when interlocutors finished discussing the first Mismatch and moved on to the remainder of the task) as a cut-off point for the early stages of dialogue. There are two main reasons for this choice. First, as we have shown in related work, through a detailed analysis of all turn transitions directly following the introduction of matching vs. mismatching landmarks, that there was a consistent and distinct reaction to the detection of the first mismatch in particular,, in both groups (in the form of longer gaps; see [25], Chapter 12.3; [46]). Essentially, the first Mismatch can thus be seen as a turning point in the interaction. Before detection of the first Mismatch, participants might feel that they are expected to give their individual contribution to the solution of a known problem (i.e. draw a path on an otherwise identical map). After the first Mismatch isit is detected, participants might feel that they need to give a joint contribution to navigate an unknown problem (i.e. the two maps are not identical), and this difference in the nature of the conversational goal can be expected to generate a difference in the interaction.

The second reason for using detection rather than resolution is that the formerlatter is far less more problematic as a time stamp also from a practical perspective. The time it took to resolve the first Mismatch varied widely across dyads (ranging in duration from under 10 seconds to over 5 minutes; for more detail see [25], Chapter 12). Moreover, even determining when a Mismatch was in fact “resolved” can be rather difficult and involves a degree of subjective judgement. In contrast, the detection of the first Mismatch was in almost all cases unambiguously expressed directly in the speech of both interlocutors. 

To conclusively examine the appropriateness of using detection of the first Mismatch as the cut-off point, we performed therefore, in addition to the results reported above, a) compared dialogue before and after resolution (rather than detection) of the first Mismatch, b) took a further analysis taking into account the three-way distinction of 1) dialogue from the start of the task to the detection of the first Mismatch, 2) dialogue during the discussion and up to the resolution of the first Mismatch and 3) all remaining (following) dialogue, and bc) used performed a separate, fully context-independent cut-off point in a continuous analysis of FTO values in the first 100 turn transitions.

Briefly, considering a three-way distinction of dialogue stages, theour analysis with a three-way distinction of dialogue stages confirmeds that we found equivalently robust evidence for a difference between the beginning and remainder of dialogues also when using resolution rather than detection of mismatches in a two-way distinction, and that there wasthere was a robust robust between-group difference only before detection, not during and after the discussion of the first Mismatch when considering ain a three-way distinction of dialogue stages (details of statistical modelling are reported in the following section).

Finally, for examining a context-independent cut-off point in a continuous analysis of mean FTO values for all transitions, we used the 40th turn transition (10% of the 400 turn transitions contained in an average dialogue on average; further details in the OSF repository) to divide conversations into an early stage and a remainder. , Fig 8 shows not only that the 40th turn transition happens to correspond quite closely to the average time point of first mismatch detection, but also that average FTO values in the ASD group tended to continuously decreased from the start of conversations until the point when the first mismatch iswas detected, strengthening the validity of using mismatch detection as a cut-off point. Note that Fig 8 shows only the first 100 turn transitions; dialogues contained a total of 400 transitions on average..

Fig 8. FTO values by turn transition and group. Positive values represent gaps; negative values represent overlaps. ASD group in blue, CTR group in green. Thin blue/green lines represent averaged FTO values by transition and group; thick lines represent fitted LOESS-smoothed curves by group, the surrounding grey shaded areas the respective standard error. The dashed vertical lines show 1) transition no. 38 (average time point for detection of first Mismatch) and, 2) transition no. 40 (context-independent cut-off; see text) and 3) transition no. 90 (average time point for resolution of first Mismatch). 

We can conclude that differences in turn-timing awere indeed greater between groups in the early stages of dialogue compared to the remainder, independent of the specific cut-off point.., regardless of the specific cut-off point used to define epochs. 

(…)

Para at l.552 needs references (at least point a) is not universally accepted, as indeed the authors seem to entertain in l.572).

Thank you for pointing out the lack of supporting references here. We have added various references and also changed the wording to mitigate the relevant statements.

These specific differences should, however, not overshadow the general finding that, at a global level, no robust differences were found in conversations between autistic as opposed to non-autistic adults. This might be considered surprising given that a) it has been shown at length in previous work that achieving rapid and precise turn-timing is highly challenging cognitively, as it can only be achieved if speakers are able to accurately predict the communicative intentions of their interlocutor [60–65], and b) predicting the behaviour of others is a skill that many autistic individuals seem to struggle with [66].

(…)

60. Bögels S, Torreira F. Listeners use intonational phrase boundaries to project turn ends in spoken interaction. J Phon. 2015;52: 46–57. 

61. De Ruiter J-P, Mitterer H, Enfield NJ. Projecting the end of a speaker’s turn: A cognitive cornerstone of conversation. Language. 2006;82: 515–535. 

62. Barthel M, Meyer AS, Levinson SC. Next speakers plan their turn early and speak after turn-final “go-signals.” Front Psychol. 2017;8: 393. 

63. Barthel M, Sauppe S, Levinson SC, Meyer AS. The timing of utterance planning in task-oriented dialogue: Evidence from a novel list-completion paradigm. Front Psychol. 2016;7: 1858. 

64. Gleitman LR, January D, Nappa R, Trueswell JC. On the give and take between event apprehension and utterance formulation. J Mem Lang. 2007;57: 544–569. 

65. Wesseling W, Son RJ van. Early preparation of experimentally elicited minimal responses. 6th SIGdial Workshop on Discourse and Dialogue. 2005. 

66. Cannon J, O’Brien AM, Bungert L, Sinha P. Prediction in Autism Spectrum Disorder: A Systematic Review of Empirical Evidence. Autism Res. 2021;14: 604–630. doi:10.1002/aur.2482

The assertion at l.565 "although it is likely that this process is still relatively more effortful for autistic individuals" seems to come from nowhere. Is it? Why? 

This is a good point—in fairness this is a speculative statement at this point, and until it can be borne out empirically (it may not be), we deem it best to not comment further. The phrase has therefore been deleted.

Our findings clearly show that at least the kinds of relatively socially motivated and high-functioning autistic adults we investigated, and at least when conversing in disposition-matched (ASD–ASD) dyads, are perfectly able to produce turn-timing of the same speed and precision as has been described for conversations between adults without a diagnosis of Autism Spectrum Disorder—although it is likely that this process is still relatively more effortful for autistic individuals..

l.642 needs citations -- what previous work? -- At least the work I am aware of on repair suggests that this usually occurs extremely locally and rapidly -- more extensive misunderstandings (that only get noticed further downstream) may result in long gaps but this does not seem to be generally true.

Thank you for this comment. We made the statement more precise and added further references here in addition to a mention earlier in the text (l. 726 in the revised manuscript with tracked changes) (copied from line 847):

We also have to acknowledge that we cannot say for sure whether long gaps, produced by any group of speakers, are appropriate or not in a given context without conducting a comprehensive qualitative analysis that takes into account the context of all turn transitions. Previous work assures us, for instance, that long gaps are typical and expected in the direct context of verbal exchanges as reactions involving misunderstanding or non-alignment. [58,80,81].

(…)

58. Kendrick KH, Torreira F. The timing and construction of preference: A quantitative study. Discourse Process. 2015;52: 255–289. 

(…)

80. Kendrick KH. The intersection of turn-taking and repair: the timing of other-initiations of repair in conversation. Front Psychol. 2015;6: 10–3389. 

81. Roberts F, Francis AL. Identifying a temporal threshold of tolerance for silent gaps after requests. J Acoust Soc Am. 2013;133: EL471–EL477. 

Reviewer 2

I think this paper makes a very important and clear contribution to the literature on turn-taking, in both special and typical populations. In brief, the authors have brought an unusual level of clarity and rigor to this topic, and they have framed their study in a way that I believe will influence future work in a positive way. I also think the limitations and caveats that they authors highlight are the right ones. All of this said, and given the paper's overall brevity and concision, I do not see a reason to not accept this paper -- I myself have no requests to make in a revision.

We are extremely grateful for the time that was taken to review and assess our paper, and for your very encouraging comments.

---

## [Editor Report · Decision Letter 1]

16 Mar 2023

PONE-D-22-26895R1Turn-timing in conversations between autistic adults: typical short-gap transitions are preferred, but not achieved instantlyPLOS ONE

Dear Dr. Wehrle, Thank you for submitting your manuscript to PLOS ONE. After careful consideration, we feel that it still requires some minor adjustment in order to meet PLOS ONE’s publication criteria. Therefore, we invite you to submit a revised version of the manuscript that addresses the points raised during the review process. The paper is clear and easy to follow. The analyses are better illustrated and generally all comments have been satisfactorily addressed. There are a few remaining points which should be fixed (see "Additional Editor Comments" below). One point concerns a piece of information which is still missing in the description of your stats and the other concerns the notion of coordinated conversational rhythm which is in this work remiains vague and which I strongly advise to keep out of your manuscript. Finally, there are a few textual comments that you should consider. All in all, the manuscript is quite ready for publication so I anticipate that if no further complexity arises, the next submission may be processed in a couple of days.

We look forward to receiving your revised manuscript.

Kind regards,

Leonardo Lancia, Ph.D

Academic Editor

PLOS ONE

Journal Requirements:

**Additional Editor Comments:**

Statistics:

You should provide some information on how you choose the dispersion parameters in the prior distributions of your models (and of course I was not suggesting to use the posteriors of one model as priors for the other one ).

Claims about coordination and conversational rhythm

p. 26, line 614: “autistic speaker pairs achieved a tightly coordinated conversational rhythm”

p.32 line 759: “The fact that ASD dyads took considerably longer to achieve typically rapid turn-timing might signify a delay in the establishment of a shared rhythm between interlocutors”.

These claims require quite a bit of supporting information and evidence which are not provided in the manuscript: how do you define conversational rhythm? Is the gap-length a reliable measure of coordination? So far only timing between turns offsets and onsets has been analyzed and discussed. Since timing and rhythm, while being related, are completely different notions, I would suggest to limit your discussion to timing issues. Alternatively, you will have to introduce the relevant notions and discuss your work in the context of the relevant literature on speech rhythm and coordination in general and on conversational rhythms in particular (and show that something like that proposed in Wilson and Wilson, 2005 actually exists).

p.4, line 93 and below: you should resolve the ICD acronym (e.g. p. 4, line 93).

p.18, line 413:  not shure about that, but you may want to substitute “signifies” with “means”.

p.19, line 442 -444: you may want to smooth the following passage. “First, as we have shown in related work through a detailed analysis of all turn transitions directly following the introduction of matching vs. mismatching landmarks, there was a consistent and distinct reaction to the detection of the first mismatch in particular, in both groups (in the form of longer gaps; see [25], Chapter 12.3; [46]).”

p.32, line 738 group -> groups

p.32 line 744 “Generally speaking, using such long gaps is an effective and successful strategy for navigating challenging and unusual situations”. This is a quite suggestive proposal but to make this claim you should be able be able to relate some communicative efficiency measure to length of these gaps.

---

## [Author Response · Author response to Decision Letter 1]

17 Mar 2023

!

PLEASE SEE ONLY THE ATTACHED PDF FOR PROPER FORMATTING

!

Copied from there:

Dear Leonardo Lancia,

Thank you kindly for your additional comments and the continued consideration of our manuscript.

We respond to all comments point-by-point below (each starting on a new page, except for the textual comments). Comments by the editor are copied in italics and smaller font, our responses are interspersed in regular type and excerpts from the revised manuscript (with tracked changes) are indented.

Please let us know if there are any further questions. 

Best regards

Simon Wehrle, Cologne, 17 March 2023

(also on behalf of my co-authors)

Editor’s comments

Statistics:

You should provide some information on how you choose the dispersion parameters in the prior distributions of your models (and of course I was not suggesting to use the posteriors of one model as priors for the other one ).

Thank you for this comment. We added an additional sentence to the manuscript to clarify. We maintain, however, that a more extensive elaboration on statistical details would detract from the readability of the paper, especially in light of the fact that we follow common practices here and provide references for the interested reader (e.g. here Lemoine, 2019), besides providing all data, scripts and model specifications for independent corroboration. 

As a specific example, for FTO we chose a prior for the slope term with a mean of 0 and a standard deviation of 1000 (milliseconds). The mean value of 0 makes the model more conservative, as the fewer data points there are, the higher the likelihood of a null result becomes. The standard deviation of 1000 is used to capture the range within which the difference between the ASD and the CTR group might fall (the very limited related evidence would suggest differences that should easily be within this range).

For the prior of the intercept, we used a mean of 0 and a standard deviation of 6000. This is intended to capture the feasible range of FTO values based on the literature: concretely, this way the model “expects” almost all turn transition values in the data set to fall between the (rather extreme) values of -12000 and +12000 milliseconds FTO.

Perhaps most importantly, as can be verified by using the code and data provided in the accompanying repository, our data set contains a high enough number of observations that any changes to the specific priors as detailed above will have (at most!) a negligible impact on the outcome of modelling in the form of the posterior distribution.

We used regularising weakly informative priors for all models [33,43]; all priors were centred at zero and distributions were chosen according to relevant results in the previous literature (e.g. the feasible range of FTO values). andWe performed posterior predictive checks with the packages brms (version 2.16.3) [40] and bayesplot (version 1.8.1) [44] in order to verify that the priors were suited to the data set. Unless otherwise specified, four sampling chains ran for 4000 iterations with a warm-up period of 2000 iterations for each model. 

 

Claims about coordination and conversational rhythm

p. 26, line 614: “autistic speaker pairs achieved a tightly coordinated conversational rhythm”

p.32 line 759: “The fact that ASD dyads took considerably longer to achieve typically rapid turn-timing might signify a delay in the establishment of a shared rhythm between interlocutors”.

These claims require quite a bit of supporting information and evidence which are not provided in the manuscript: how do you define conversational rhythm? Is the gap-length a reliable measure of coordination? So far only timing between turns offsets and onsets has been analyzed and discussed. Since timing and rhythm, while being related, are completely different notions, I would suggest to limit your discussion to timing issues. Alternatively, you will have to introduce the relevant notions and discuss your work in the context of the relevant literature on speech rhythm and coordination in general and on conversational rhythms in particular (and show that something like that proposed in Wilson and Wilson, 2005 actually exists).

Thank you. Good point, we agree that we cannot confidently make specific claims about coordination or rhythm within the scope of the analyses presented in this paper and would not want to speculate too widely here. We have revised the manuscript accordingly (also in one additional place where “rhythm” had been mentioned). Copied from lines 431, 615 and 763, respectively:

This variability disappeared after the initial stages, as the dyads seemed to seemingly settled into a temporally stable turn-taking stylestable rhythm of turn-timing, one whichthat is virtually indistinguishable from that of CTR dyads. 

(…)

Our finding that differences in turn-taking between groups, in the form of longer gaps in the conversations of autistic dyads, were found only in the earliest stages of dialogue can be interpreted suchshows that autistic speaker pairs successfully achieved established a tightly coordinated conversational rhythmdegree of rapid turn-timing, that is essentially indistinguishable from that of non-autistic dyads, but that they did not do so instantly [cf. 57].

(…)

The fact thatWhile ASD dyads took considerably longer to achieve establish a typicallytypically rapid exchange of turns turn-timing might signify a delay in the establishment of a shared rhythm between interlocutors. However, such a this deviation from the superficially equivalent behaviour between groups at the a global level may be functionally motivated, and may indeed represent the most appropriate behaviour for this group of speakers in the early stages of a social interaction.

 

p.4, line 93 and below: you should resolve the ICD acronym (e.g. p. 4, line 93).

The key diagnostic criteria described in the ICD-10 (International Statistical Classification of Diseases and Related Health Problems) are 1) unusual (“impaired”) social interaction and communication and 2) a restricted repertoire of activities and interests.

p.18, line 413: not shure about that, but you may want to substitute “signifies” with “means”.

We respectfully disagree and would prefer to keep the original formulation. The only issue I might see is a confusion with “significance” in the statistical sense, but I don’t think this is a concern in the case of the specific formulation (“signifies”) used here.

p.19, line 442 -444: you may want to smooth the following passage. “First, as we have shown in related work through a detailed analysis of all turn transitions directly following the introduction of matching vs. mismatching landmarks, there was a consistent and distinct reaction to the detection of the first mismatch in particular, in both groups (in the form of longer gaps; see [25], Chapter 12.3; [46]).”

First, as we have shown in related work, through a detailed analysis of all turn transitions directly following the introduction of matching vs. mismatching landmarks, that there was a consistent and distinct reaction to the detection of the first mismatch in particular, in both groups (in the form of longer gaps; see [25], Chapter 12.3; [46]).

p.32, line 738 group -> groups

We have resolved this in a slightly different way (and “group” was in fact not grammatically incorrect in the original formulation):

Conversely, we have shown in related work on the same data that dyads from the CTR and the ASD group produced equally long gaps immediately following the introduction of mismatching landmarks, but that dyads from the ASD group produced longer gaps immediately following the introduction of matching landmarks thancompared to the CTR group

p.32 line 744 “Generally speaking, using such long gaps is an effective and successful strategy for navigating challenging and unusual situations”. This is a quite suggestive proposal but to make this claim you should be able be able to relate some communicative efficiency measure to length of these gaps.

We changed the formulation to reflect your comment and the fact that this is not an empirical fact based on the analyses presented in the paper:

). Generally speaking, using such long gaps may beis an effective and successful strategy for navigating challenging and unusual situations, and it is employed by both groups of speakers in our data.

---

## [Editor Report · Decision Letter 2]

22 Mar 2023

Turn-timing in conversations between autistic adults: typical short-gap transitions are preferred, but not achieved instantly

PONE-D-22-26895R2

Dear Dr. Wehrle,

We’re pleased to inform you that your manuscript has been judged scientifically suitable for publication and will be formally accepted for publication once it meets all outstanding technical requirements.

Kind regards,

Leonardo Lancia, Ph.D

Academic Editor

PLOS ONE
---

## [Editor Report · Acceptance letter]

28 Mar 2023

PONE-D-22-26895R2 

Turn-timing in conversations between autistic adults: typical short-gap transitions are preferred, but not achieved instantly 

Dear Dr. Wehrle:

I'm pleased to inform you that your manuscript has been deemed suitable for publication in PLOS ONE. Congratulations! Your manuscript is now with our production department. 

Kind regards, 

on behalf of

Dr. Leonardo Lancia 

Academic Editor

PLOS ONE